# Making A Trade-Off Between Cost and Distance By A Differentiable Way

## Abstract

The Cost-Distance problem, introduced by Meyerson, which is a natural abstraction for modeling UAV logistics networks, seeks a network design that simultaneously minimizes construction cost and the weighted routing distances from multiple sources to a designated root. Existing methods exhibit a strong dependence on the number of sources and are difficult to parallelize, which hinders their scalability on large graphs. We propose Cost-Distance Policy Gradient (CDPG), the first gradient-based framework for this problem. CDPG relaxes the discrete subgraph selection into a probabilistic adjacency matrix and formulates the Cost-Distance objective as an expectation, enabling efficient optimization via policy gradients. Our algorithm achieves the time complexity of $\mathcal{O}(m \log n)$, faster than the previous fastest approximation algorithm's $\mathcal{O}(|S|(m+n \log n))$ in graphs with dense sources. Extensive experiments across 9 real-world Unmanned Aerial Vehicle (UAV) logistics scenarios in the Guangdong-Hong Kong-Macao Greater Bay Area demonstrate that CDPG significantly outperforms approximation algorithms, continuous relaxation baselines, and heuristic search methods. Our code is available at: `https://anonymous.4open.science/r/iclr_cdpg-8737`.

## 1 Introduction

The rapid development of the Low-Altitude Economy (LAE) has significantly altered the landscape of urban transportation, with unmanned aerial vehicles (UAVs) playing a pivotal role in shaping the logistics networks of the future. As UAV technology continues to evolve, the planning and optimization of UAV logistics networks has become an increasingly important area of research.

There are many studies in logistics network planning, such as the Hub Location and Routing Problem (HLRP) Wu et al. (2024) and Vehicle Routing Problem (VRP) Gao et al. (2022). However, they deviate from UAV-based low-altitude logistics, where UAVs have small payloads and large fleet sizes, and need to be modeled as flows. Existing works on UAV network planning Feng & Cao (2025); Rashid Nazir et al. (2025); Qi et al. (2025) commonly account for both urban airspace construction costs and UAV flow routing distance.

Meyerson et al. (2008) introduced the **Cost-Distance problem**, abstracting the above shared characteristics into a unified framework. Specifically, it optimizes both the construction cost of a network and the total weighted routing distances between a set of sources and a designated root.

Formally, consider a connected graph $\mathcal{G} := (\mathcal{V}, \mathcal{E})$ where $\mathcal{V}$ is the set of $n := |\mathcal{V}|$ nodes and $\mathcal{E}$ is the set of $m := |\mathcal{E}|$ edges. The topology of the graph is specified by an adjacency matrix $X \in \{0,1\}^{n \times n}$. Each edge has a cost weight $C \in \mathbb{R}_{>0}^{n \times n}$ and a distance weight $D \in \mathbb{R}_{>0}^{n \times n}$. Given a root node $r$ and a set of source nodes $S \subseteq \mathcal{V}$, the goal of the Cost-Distance Problem is to select an adjacency matrix $X'$ of a connected subgraph $\mathcal{G}' := (\mathcal{V}', \mathcal{E}')$, where $S \subseteq \mathcal{V}' \subseteq \mathcal{V}$ and $\mathcal{E}' \subseteq \mathcal{E}$, that minimizes the joint objective:

$$J(X') := \underbrace{\sum_{i,j} X'_{ij} C_{ij}}_{\Omega_C(X')} + \underbrace{\sum_i w_i L_D^{X'}(v_i, r)}_{\Omega_D(X')}. \tag{1}$$

$w \in \mathbb{R}_{\geq 0}^n$ is a node weight. $w_i > 0$ if and only if $v_i \in S$, and $w_i = 0$ otherwise. $L_D^{X'}(v, r) : \mathcal{V} \times \mathcal{V} \to \mathbb{R}_{\geq 0}$ denotes the shortest-path distance from $v$ to $r$ in the selected subgraph under selected edge $X'$

weight $D$. The objective can be decomposed into two terms. $\Omega_C(X')$ is the construction cost, which measures the total cost of the selected edges. In contrast, $\Omega_D(X')$ is the routing distance, representing the weighted sum of the shortest-path distances. This problem formulation directly models the real-world UAV logistics scenario, the root node $r$ represents the central warehouse, source nodes $S$ are UAV locations, edges represent feasible low-altitude UAV air corridors, the remaining nodes correspond to charging stations acting as Steiner points and node weight $w_i$ denotes UAV flow.

This tension between construction cost and routing distance appears in many combinatorial optimization problems. For example, in the Survivable Network Design Problem (SNDP), adding redundant links improves connectivity and reduces routing distances but raises construction expenses Gabow et al. (1998); Gupta et al. (2009). The Shallow-Light Steiner Tree Chen & Young (2019) and the Buy-at-Bulk network design Guha et al. (2009) are special cases of the Cost-Distance problem, making it a fundamental model for cost-sensitive routing and infrastructure optimization.

Since its introduction, there has been significant progress on approximation algorithms for this problem. Meyerson et al. (2008) proposed the first randomized hierarchical aggregation algorithm with an $\mathcal{O}(\log|S|)$-approximation ratio and a running time of $\mathcal{O}(|S|^2(m + n\log n))$. Chekuri et al. (2001) later provided a derandomized version using a linear-programming dual construction, making the algorithm more controllable. More recently, Held & Perner (2025) showed that the running time can be improved to $\mathcal{O}(|S|(m + n\log n))$, which is the most advanced method for this problem. On the hardness side, Chuzhoy et al. (2008) proved that this problem cannot be approximated better than $\Omega(\log\log n)$ unless $\mathrm{NP} \subseteq \mathrm{DTIME}\big(n^{\mathcal{O}(\log\log\log n)}\big)$.

From time complexity and the approximation factor, it is evident that the running time scales linearly with $|S|$, and when $|S| = \mathcal{O}(n)$, the approximation factor becomes several times away from the optimum. However, in the UAV network, sources are often dense as UAVs frequently recharge or transfer at delivery stations with flow demands. Moreover, the parallelism of existing algorithms is rather poor. Therefore, we focus on: whether more parallelized and more efficient algorithms exist for dense-source settings and whether algorithms can yield higher-quality solutions.

In recent years, machine learning methods have been explored for combinatorial optimization. In the supervised setting Gasse et al. (2019); Gupta et al. (2020), models learn from labeled instances for fast inference on unseen data, but such labels are costly to obtain, and the Cost-Distance problem has no labeled dataset. In the unsupervised setting, reinforcement learning (RL)-based approaches Khalil et al. (2017); Li et al. (2021) avoid labels but still rely on exploring rewards and expensive episodic training. If instead we could derive a continuous relaxation that encodes both the objective and constraints into an efficiently parallelized differentiable loss, it would enable: (i) gradient-based optimization on a single instance, and (ii) highly parallelized processing of the problem on GPUs.

We propose **Cost-Distance Policy Gradient (CDPG)**, the first framework providing a continuous relaxation for Cost-Distance optimization, which consists of two components: **Bilinear Edge Policy (BEP)** and the **Value Solver (VS)**. The former parameterizes the solution $X'$ as the decision variable, while the latter evaluates the corresponding Cost-Distance objective induced by the policy. Value Solver relaxes the construction cost and the routing distance respectively. The construction cost term has been widely studied, and relaxing edge selection variables is relatively straightforward Stewart et al. (2023). However, for routing distance term, existing work relaxes the edge weights rather than the edge selection itself such as DataSP Lahoud et al. (2024) and gradient projection algorithm based on the Goldstein–Levitin–Polyak method Wu & Huang (2014), which cannot solve our problem.

To relax routing distance differentially, we formulate the route decision as a Markov Decision Process (MDP) Puterman (1990), treating edge selection as a stochastic policy and computing the expected routing distance via the Bellman equation. This yields a differentiable mapping from the decision variables to the expected path length. However, it suffers from two fundamental difficulties. First, graphs inevitably contain cycles, the Bellman equation oscillates on loops and requires many iterations to approximately converge, leading to prohibitively high time complexity. Second, even after convergence, the resulting solution cannot guarantee a valid directed arborescence distribution; after rounding, the solution quality degrades significantly.

To address these challenges, we design an **acyclic mask** in BEP tailored for the Cost-Distance problem. By enforcing a topological order during optimization, the policy distribution is directly

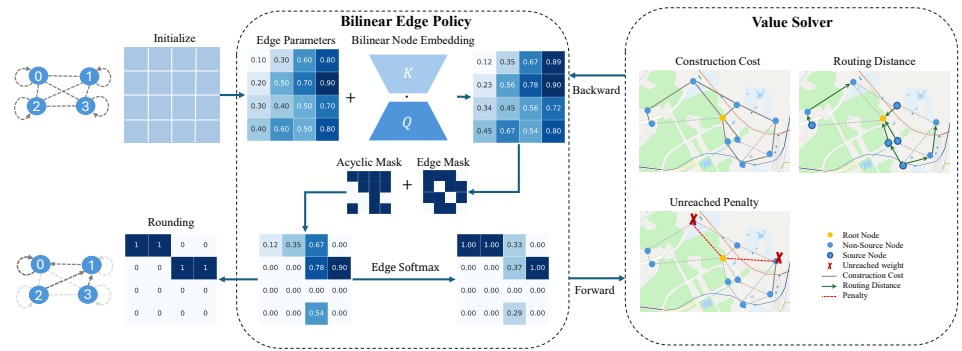

Figure 1: This figure shows the framework of CDPG, including its main components: Bilinear Edge Policy and Value Solver.

restricted to a valid acyclic space. Furthermore, to accelerate convergence, we set a truncation of Neumann iteration when solving the Bellman equation and estimate the remaining values by **unreached penalty**. Lastly, BEP not only parameterizes edge selection with edge logits, but also takes a **bilinear node embedding**, to capture low-rank structure and help escape poor local minima.

Since there are no existing open-source Cost-Distance datasets, we obtain a real-world UAV logistics network in the Guangdong-Hong Kong-Macao Greater Bay Area to evaluate CDPG. Our main contributions are summarized as follows:

- We propose CDPG, the first gradient-based framework for the Cost–Distance problem, formulating the objective as an expectation over a probabilistic adjacency matrix.
- We theoretically analyze CDPG and prove that CDPG achieves a total runtime of $\mathcal{O}(m \log n)$. It improves over the fastest existing approximation algorithm with complexity $\mathcal{O}(|S|(m+n \log n))$ when $|S| > \min\{\log n, m/n\}$. This implies that CDPG underperforms solely in severely sparse source node graphs. Moreover, CDPG can be efficiently parallelized and deployed on GPUs, further enhancing its practical scalability.
- We conduct extensive experiments on 9 real-world UAV logistics scenarios in the Guangdong-Hong Kong-Macao Greater Bay Area, showing that CDPG significantly outperforms the approximation algorithm, continuous relaxation baselines, and heuristic methods.

## 2 METHODOLOGY

### 2.1 FRAMEWORK OVERVIEW

To optimize the Cost-Distance Problem, we propose the CDPG algorithm, whose framework is illustrated in Figure 1. We first perform a probabilistic relaxation of the original problem as a preparation for its subsequent continuous formulation.

Then we propose **Bilinear Edge Policy**. BEP directly parameterizes the probabilistic matrix as a combination of learnable edge logits and bilinear node embeddings with edge and acyclic mask.

Given the policy, we design **Value Solver** to estimate the expected objective. To make the routing distance objective differentiable in a continuous manner, we, for the first time, model the routing distance as a Markov Decision Process (MDP) in a combinatorial problem. This formulation allows the use of the Bellman equation to guarantee differentiability. Therefore, we can optimize the BEP using gradient-based methods such as Adam Kingma (2014), Adagrad Duchi et al. (2011), and so on. The cost gradients are backpropagated to update the BEP iteratively.

Once training is completed, we select, for each source node, the edge with the highest probability as a selected path to the parent node. A non-source node will only be assigned a parent node if there is a route from the source node passing through it. This ensures that the final output is an arborescence. The whole algorithm of CDPG is shown in Appendix C.

## 2.2 PROBLEM RELAXATION

We transform the discrete solution $X'$ into a probabilistic matrix $P \in [0,1]^{n \times n}$ formulation and rewrite the objective as its expectation, a process known as probabilistic relaxation of the original problem. Since previous work has proved that the optimal solution of the Cost-Distance Problem is an arborescence, we further impose structural constraints (3) and (4) on the relaxed problem to ensure that the resulting solution remains an arborescence, thereby reducing the solution space and improving optimization efficiency.

$$\min_P \mathbb{E}_{X' \sim P}[J(X')] := \mathbb{E}[\Omega_C] + \mathbb{E}[\Omega_D] \tag{2}$$

$$\text{s.t. } \sum_j P_{rj} = 0, \forall e_{rj} \in \mathcal{E}, \quad \sum_j P_{ij} \begin{cases} = 1 & \text{if } v_i \in S, \\ \leq 1 & \text{otherwise,} \end{cases}, \tag{3}$$

$$\text{tr}\left(P^k\right) = 0, \quad \forall k \in \mathbb{N}^+. \tag{4}$$

Equation (3) ensures that the root node has no parent node and each source node has a unique parent, while a non-source node can have 0 or 1 parent. Equation (4) guarantees that $P$ is acyclic.

## 2.3 BILINEAR EDGE POLICY

Bilinear Edge Policy parameterized $P$ as $P^\theta \in \mathbb{R}^{n \times n}$. The design of $P^\theta$ incorporates the following components:

**1. Optimizable Parameters:** The matrix $P^\theta$ is parameterized as $P^\theta = H + l^{-1/2} Q^\top K, \theta = \{H, Q, K\}$, where $H \in \mathbb{R}^{n \times n}$ and $Q, K \in \mathbb{R}^{l \times n}, l \ll n$. $K$ and $Q$ are initialized using a standard Gaussian distribution $\mathcal{N}(0, 1)$, and then normalized $\|Q\|_2 = \|K\|_2 = 1$ during optimizing. Our bilinear node embedding is inspired by VGAE Kipf & Welling (2016), where adjacency is reconstructed via inner products of node embeddings. We extend this to directed graphs by assigning each node two embeddings, capturing low-rank structure and helping escape poor local minima in optimization. To retain high-rank expressiveness, we also preserve $H$.

**2. Edge Mask and Acyclic Mask:** $X$ is applied to ensure only the edges present in $\mathcal{G}$ can have non-zero probabilities. Another binary mask $A$ is incorporated to eliminate cycles. This mask is derived based on a new distance metric from each node to a central node in the graph:

$$P^\theta_{\text{mask}} := P^\theta \odot X \odot A, \ A_{ij} := \begin{cases} 1 & L^X_\Phi(v_i, r) > L^X_\Phi(v_j, r), \\ 0 & \text{otherwise.} \end{cases} \tag{5}$$

We design $\Phi \in \mathbb{R}^{n \times n}$ as a new distance metric tailored for the Cost-Distance problem: $\Phi = (W + \varepsilon I)^{\odot -1} \odot C + D$. Specifically, for each source node $s$, its associated weight $w_s$ is fully assigned along its shortest path to the root node $r$, and $W_{ij}$ is the total amount of weight passing through edge $(i, j)$ over all such source-root paths. $\varepsilon > 0$ is a small positive constant to avoid division by zero. By computing $\Phi$ for all edges, we derive a topological ordering that reflects their relative structural priorities. Based on this ordering, we construct an acyclic mask on the original graph to ensure valid structural constraints.

**Theorem 1** (Optimal Distance metric). *Let $W^* \in \mathbb{R}^{n \times n}$ denote the edge weight matrix induced by the optimal solution of the Cost-Distance problem. Then $W^*$ satisfies*

$$W^* = \text{Dijkstra}(\Phi^*, r), \quad \Phi^* = (W^* + \varepsilon I)^{\odot -1} \odot C + D.$$

*This theorem is proved in Appendix B.1.*

Since directly computing $W^*$ is NP-hard (equals to solving Cost-Distance problem), we approximate it by a constant matrix: $W := \|w\|_1/(2n) \cdot \mathbf{1}_{n \times n}$. The exact solution is subsequently refined through gradient-based optimization, allowing the mask to guide acyclicity without requiring precise initial weights.

**Discussion (Comparison with soft DAG penalty)**: Many works treat acyclic constraint as a soft DAG penalty, such as NOTEARS Zheng et al. (2018), DAG-GNN Yu et al. (2019), and Nocurl Yu et al. (2021). They promote acyclicity via a penalty with a computational complexity of $\mathcal{O}(n^3)$, leading to slow optimization. Moreover, it only reduces cycles rather than strictly eliminating them. In contrast, our acyclic mask derives a topological ordering to construct an exact acyclic mask, ensuring a strict acyclic structure with much lower computational cost.

**3. Edge Normalization and Rounding:** Finally, the edge probabilities for each node are normalized using an edge-softmax function:

$$P^\theta_{\text{norm},ij} := \frac{\exp(P^\theta_{\text{mask},ij})}{\sum_{k \in \mathcal{V}_{\text{mask}}(i)} \exp(P^\theta_{ik})}, \quad i \neq r, \tag{6}$$

where $\mathcal{V}_{\text{mask}}(i)$ represents the neighbors of node $i$ in the masked graph. Here, $P^\theta_{\text{norm}}$ equals to probabilistic matrix $P$. After BEP finishes optimizing, we use the edge-argmax function to round $P^\theta$:

$$\hat{X}'_{ij} := \begin{cases} 1 & \text{if } P_{ij} = \max\{P_{ik} | k \in \mathcal{V}_{\text{mask}}(i)\} \\ 0 & \text{otherwise.} \end{cases} \tag{7}$$

Finally, we prune non-source nodes through which no weight passes in $\hat{X}'$ to get the solution $X'$.

## 2.4 VALUE SOLVER

Value Solver estimates the Cost-Distance objective by calculating $\mathbb{E}[\Omega_C]$ and $\mathbb{E}[\Omega_D]$ in a differentiable and efficient way. First it is simple to calculate $\mathbb{E}[\Omega_C] = \text{tr}\left(P^\top C\right)$. Then, to estimate $\mathbb{E}[\Omega_D]$, we treat the policy matrix $P$ as the decision process of MDP where the reward is coupled with distance. Under this formulation, the routing distance term can be reformulated as a cumulative reward maximization problem. The main components of the process are detailed as follows.

- **State:** The state $s_t$ is index of node in round $t$.
- **Action:** Given the state $s_t$, the agent decides an action $a_t \in \mathcal{A}_{s_t}$ following the policy $\pi(a_t|s_t)$, where $\mathcal{A}_{s_t}$ is set of outgoing paths which from $s_t$.
- **State transition:** Under the state $s_t$ taking action $a_t$, node weight will transfer to next state $s_{t+1}$ reached by $a_t$.
- **Reward:** While transferring from $s_t = u$ to $s_{t+1} = v$, we use distance to define the reward. $r(s_t, a_t) := -D_{uv}$. And the expectation of the immediate reward of state $s_t$ given policy $\pi$ satisfies

$$\mathbb{E}_{a_t \sim \pi(\cdot|s_t)}[r(s_t)] = \sum_{a_t \in \mathcal{A}} \pi(a_t|s_t) r(s_t, a_t). \tag{8}$$

Under the definition of MDP, the discounted return is formulated as $R(s_i) = \sum_{t=1}^{t'} \gamma^{t-1} r(s_{i+t}, a_{i+t})$. $\gamma \in (0,1]$ represents the discount factor and $s_{i+t'}$ is the root node. We use value function $V_\gamma(\cdot) : \mathcal{S} \rightarrow \mathbb{R}$ to denote the expectation discounted return of each state: $V_\gamma(s) = \mathbb{E}[R(s)]$. Thus, our object can be evaluated as $\mathbb{E}[\Omega_D] = -w^\top \mathbf{V}_1(s)$ where $\mathbf{V}_\gamma(\cdot) : \mathcal{S}^n \rightarrow \mathbb{R}^n$ is a vector form of $V$. The key to calculating $\Omega_D$ is to estimate $\mathbf{V}$.

**Theorem 2.** *According to the Bellman equation, the vector value for a given policy table $P^\theta$ satisfies*

$$\mathbf{V}_\gamma(\mathbf{s}) = -(I - \gamma P)^{-1}(P \odot D)\mathbf{1}_n. \tag{9}$$

*where $\mathbf{1}_n$ denotes a vector of ones with dimension $n$. What's more, because $P^\theta$ is acyclic, $(I - \gamma P^\theta)^{-1}$ is invertible when $\gamma = 1$, making $\mathbf{V}$ an unbiased estimation of routing distance. This theorem is proved in Appendix B.2.*

However, it's hard to estimate $\mathbf{V}$ through Theorem 2, since computation of matrix inversion requires a time complexity of $\mathcal{O}(n^3)$. Therefore, we use Neumann iteration Greenbaum (1997) to approximate the matrix inverse, which computes a matrix inverse involving repeated matrix multiplications in Equation (10). Matrix multiplication is inherently parallelizable for GPUs, where matrix multiplication can be distributed across multiple processing units.

$$\mathbf{V}_1(\mathbf{s}) = -\underbrace{\sum_{t=1}^{+\infty} P^t \odot D\mathbf{1}_n}_{\text{cyclic}} = -\underbrace{\sum_{t=1}^{n} P^t \odot D\mathbf{1}_n}_{\text{acyclic}}. \tag{10}$$

**Theorem 3.** *During the Neumann iteration process, only $n-1$ iterations are needed to obtain the exact solution, without requiring additional iterations for approximation, since the acyclic mask prevents cycles in $P^\theta$. This theorem is proved in Appendix B.3.*

However, $n$ still grows large as the number of nodes in the graph increases. To improve the computation speed of $\mathbb{E}[\Omega_D]$, we approximate the solution by introducing a penalty term $\mu$ for unreached weight.

$$\mathbb{E}[\Omega_D] \geq \mathbb{E}[\Omega_D^{\text{eval}}] \coloneqq w^\top \sum_{t=1}^{d} P^t \odot D\mathbf{1}_n + \mu, \quad \mu \coloneqq w^\top P^d L_D^{X \odot A}(\mathcal{V}, r). \tag{11}$$

To balance computation complexity and estimation relative error in $\mathcal{O}(1)$, $d$ satisfies $d \propto \log n$ (see Theorem 5). $\mu$ means after $d$ rounds of MDP, all of the weight that has not reached the root will get unreached penalty, which is proportional to the minimum distance between the current node and the root node. Furthermore, to accelerate computation, by swapping the order of operations, we avoid matrix multiplication during the computation of $\Omega_D$, and only vector-matrix multiplication is performed. This operation can be parallelized and deployed on GPUs. The detailed calculation is shown in Algorithm 1. As a result, the computational complexity is reduced to $\mathcal{O}(md)$. Finally, we can evaluate (2) with policy parameter $\theta$ and use gradient descent with step size $\eta$ to optimize $\theta$.

$$\theta' = \{H', Q', K'\} = \{H - \eta \nabla_H \mathbb{E}[J], Q - \eta \nabla_Q \mathbb{E}[J], K - \eta \nabla_K \mathbb{E}[J]\}. \tag{12}$$

# 3 THEORETICAL ANALYSIS

In this section, we provide theoretical guarantees for the proposed CDPG framework. These theorems show that CDPG converges in time complexity $\mathcal{O}(m \log n)$.

**Theorem 4** (Lipschitz continuity of the relaxed objective. Proved in Appendix B.4). *Let $\mathbb{E}[J(\theta)]$ be defined as $\mathbb{E}[J(\theta)] = \mathbb{E}[\Omega_C] + \mathbb{E}[\Omega_D^{\text{eval}}]$. The gradient $\nabla_\theta \mathbb{E}[J]$ is Lipschitz continuous, with Lipschitz constant bounded as*

$$L \leq \sqrt{1 + 2l^{-1/2}} \left( \|w\| \|D\|_F \cdot \frac{1+\rho}{(1-\rho)^3} + \|w\| \|L_D^{X \odot A}\| \cdot \frac{4}{(1-\rho)^2 e} \right). \tag{13}$$

*where $\rho \in (0, 1)$ is defined as the maximal Frobenius contraction rate of matrix powers:*

$$\rho \coloneqq \sup_{t \geq 1} \|P^{t+1}\|_F / \|P^t\|_F. \tag{14}$$

**Theorem 5** (Estimation Relative Error bound. Proved in Appendix B.5). *The estimation relative error between $\mathbb{E}[\Omega_D^{\text{eval}}]$ and $\mathbb{E}[\Omega_D]$ is bounded by*

$$(\mathbb{E}[\Omega_D] - \mathbb{E}[\Omega_D^{\text{eval}}]) / \mathbb{E}[\Omega_D] \leq n \frac{\|D\|_{\max}}{\|D\|_{\min}} \rho^d. \tag{15}$$

*To guarantee the relative error is $\mathcal{O}(1)$, $d$ satisfies $d \propto \log n$.*

**Theorem 6** (Convergence Rate of Gradient Descent. Proved in Appendix B.6). *Consider optimizing the relaxed Cost-Distance objective $\mathbb{E}[J(\theta)]$ via gradient descent with fixed step size $\eta \leq \frac{1}{L}$, where $L$ is the gradient Lipschitz constant from Theorem 4. Let $\theta_{\text{loc}}$ denote a stationary point of $\mathbb{E}[J(\theta)]$. Then, after $T$ iterations, the suboptimality gap satisfies*

$$(\mathbb{E}[J(\theta_T)] - \mathbb{E}[J(\theta_{\text{loc}})]) \leq L \|\theta_0 - \theta_{\text{loc}}\|_F^2 / 2T. \tag{16}$$

*Note that the optimization goal is to find policy parameters $\theta$, but not necessarily to evaluate $\mathbb{E}[J]$ itself. If we scale $\mathbb{E}[J]$ by a constant positive factor $\frac{1}{L} > 0$, i.e., let $\mathbb{E}[\tilde{J}] = \frac{\mathbb{E}[J]}{L}$, then the gradient becomes $\nabla \mathbb{E}[\tilde{J}] = \frac{1}{L} \nabla \mathbb{E}[J]$, and the update rule becomes:*

$$\theta_{t+1} = \theta_t - \eta \nabla_{\theta_t} \mathbb{E}[\tilde{J}] = \theta_t - \frac{\eta}{L} \nabla_{\theta_t} \mathbb{E}[J]. \tag{17}$$

*This is equivalent to performing gradient descent on $J$ with effective step size $\tilde{\eta} = \frac{\eta}{L}$, so the iteration trajectory is preserved modulo step size rescaling.*

*Moreover, in adaptive optimizers such as Adam, even this step-size difference is absorbed through internal rescaling, so the optimizer's behavior is invariant to scalar scaling of the objective. Therefore, in our analysis and design, we may ignore constant multiplicative factors in $L$, and report iteration complexity $T = \mathcal{O}(1)$.*

**Theorem 7** (Overall Computational Complexity. Proved in Appendix B.7). *The total computational complexity of CDPG is: $\mathcal{O}(m \log n)$. And the $\mathcal{O}(m)$ part can be efficiently parallelized and deployed on GPUs.*

# 4 EXPERIMENT

In this section, we first introduce a novel dataset from real-world UAV logistics transportation scenarios. We then extensively evaluate the proposed CDPG on this dataset, benchmarking it against several approaches. Experimental results consistently confirm that CDPG achieves superior performance. To further elucidate the reasons behind its effectiveness, we also conduct comprehensive ablation studies and present insightful analyses.

## 4.1 EXPERIMENTAL SETUP

### 4.1.1 DATASET INTRODUCTION

We obtained the logistics node and edges from Tencent Location Service to construct a network model of the Greater Bay Area. The source data was publicly available online. After preprocessing, the network was partitioned into 9 subgraphs, each centered on a designated core node, with nodes assigned to the core they are geographically closest to, yielding a proximity-based decomposition. Table 3 summarizes the key statistics of 9 datasets.

We extracted three types of graph attributes: node weights, distance weights, and cost weights (see Appendix D). Node weights were derived from mobile signaling data reflecting online shopping activity; distance weights represent realistic UAV flight paths in urban settings; cost weights were computed by combining these path lengths with drone performance parameters.

### 4.1.2 EVALUATION METRICS AND BASELINES

We assessed performance using two metrics—the Cost-Distance objective $J$ and execution time. Each method was run 20 times, and the reported results are the averages over those trials.

We evaluate the performance of CDPG against several baselines. Since no labeled dataset is available, supervised learning methods are not applicable. Moreover, due to the NP-hardness of the problem, no exact algorithm is feasible for large-scale graphs. Since the states of our MDP model are simple indices and finite, we do not employ a GNN-based method to parameterize our policy.

For evaluation, we adopt Held Routing Held & Perner (2025), which, to the best of our knowledge, is the most recent and advanced approximation approach. CDPG is the first continuous relaxation for the Cost-Distance problem, and its formulation does not allow existing continuous relaxation works to be directly applied. However, two components in our framework can be replaced with classical techniques from related domains, yielding several gradient-based variants. Overall, the baselines fall into three categories, as detailed below:

- **Heuristic Search** We adopted heuristic search methods due to their widespread use in logistics network optimization. Specifically, we employed Simulated Annealing (SA) Van Laarhoven et al. (1987) and Genetic Algorithm (GA) Mitchell (1998).
- **Gradient Based Method With Continuous Relaxation** These baselines differ from CDPG in their continuous relaxation strategies, primarily in two aspects: (1) Constraint handling: While Notears Zheng et al. (2018) imposes continuous acyclicity penalties during optimization, CDPG enforces acyclicity via a discrete mask; (2) Decision paradigm (reinforcement learning). CDPG assigns weights deterministically based on a softened probability distribution, whereas methods such as Gumbel Softmax (GS) Jang et al. (2016) and Maximum Entropy Policy (MEP) Haarnoja et al. (2018) rely on stochastic sampling. Here, sampling refers to drawing decisions from a probability distribution, analogous to reinforcement learning. We named the above three baselines as CDPG-N, CDPG-MEP and CDPG-GS respectively.
- **Approximation Method** Held Routing Held & Perner (2025) employs a modified Kruskal algorithm that iteratively constructs routing trees by minimizing a hybrid cost-distance objective. The method dynamically adjusts edge weights during path exploration to balance construction cost and routing distance.

### 4.1.3 IMPLEMENTATIONS

In CDPG, we set the number of epochs to $T = 300$, which is sufficient for convergence on all datasets, i.e., when $\frac{|J(\theta_t) - J(\theta_{t-1})|}{J(\theta_t)} < 10^{-6}$. And the number of iterations in the loss computation

Table 1: Comparison of Other Methods with Ours. The evaluation metric is $J\,(\times 10^4)$. Boldface indicates the lowest (i.e., best) value in each column. An asterisk (*) denotes statistically significant improvements ($p < 0.05$) over the best baseline, and a dagger ($\dagger$) marks the best result of baselines.

| Method | Graph 1 | Graph 2 | Graph 3 | Graph 4 | Graph 5 | Graph 6 | Graph 7 | Graph 8 | Graph 9 |
|---|---|---|---|---|---|---|---|---|---|
| GA | 1248.15 | 682.50 | 72.10 | 165.68 | 172.93 | 434.73 | 158.63 | 157.74 | 88.06 |
| SA | 1163.98 | 602.65 | 66.57 | 209.93 | 139.40 | 346.01 | 155.39 | 141.85 | 86.36 |
| CDPG-N | 221.37 | 198.11 | 28.76 | $^\dagger$51.87 | $^\dagger$66.94 | $^\dagger$126.53 | $^\dagger$71.45 | 119.60 | 65.92 |
| CDPG-MEP | 216.82 | $^\dagger$188.33 | 32.33 | 61.52 | 68.07 | 132.86 | 94.06 | 133.75 | 66.81 |
| CDPG-GS | $^\dagger$215.93 | 190.30 | 32.48 | 61.34 | 69.00 | 133.74 | 94.24 | 138.30 | 68.28 |
| Held | 898.06 | 583.81 | $^\dagger$27.32 | 105.18 | 113.56 | 303.85 | 139.47 | $^\dagger$114.84 | $^\dagger$63.50 |
| CDPG | ***158.85** | ***143.73** | ***24.80** | ***44.40** | ***58.83** | ***109.94** | ***51.68** | ***100.77** | ***61.91** |
| *Improvement* | 26.43% | 23.68% | 9.22% | 14.40% | 12.12% | 13.11% | 27.67% | 12.25% | 2.50% |
| – AcyM | 204.72 | 170.63 | 29.32 | 52.00 | 66.52 | 127.01 | 67.32 | 121.56 | 68.74 |
| – URP | 157.19 | 143.20 | 24.91 | 44.63 | 59.12 | 110.31 | 51.95 | 100.76 | 61.86 |
| – BNE | 166.06 | 152.08 | 25.66 | 47.47 | 62.01 | 116.16 | 54.04 | 104.35 | 64.47 |

Table 2: Average runtime (in seconds) across different algorithms and datasets. Boldface indicates the lowest (i.e., best) runtime in each column.

| Method | Graph 1 | Graph 2 | Graph 3 | Graph 4 | Graph 5 | Graph 6 | Graph 7 | Graph 8 | Graph 9 |
|---|---|---|---|---|---|---|---|---|---|
| GA | 306.38 | 227.42 | 20.33 | 65.69 | 50.42 | 116.85 | 59.23 | 24.96 | 11.45 |
| SA | **20.75** | **14.20** | **1.97** | **4.41** | **3.57** | **7.23** | **3.90** | **2.04** | **1.40** |
| CDPG-N | 6728.69 | 5342.73 | 93.71 | 1063.81 | 614.27 | 3194.64 | 863.11 | 146.82 | 28.13 |
| CDPG-MEP | 105.16 | 70.33 | 7.33 | 19.45 | 15.15 | 34.14 | 17.57 | 8.61 | 5.28 |
| CDPG-GS | 105.41 | 70.30 | 7.31 | 19.50 | 15.16 | 34.25 | 17.50 | 8.62 | 5.25 |
| Held | 1057.83 | 610.48 | 19.87 | 119.14 | 85.22 | 259.58 | 99.71 | 24.24 | 6.17 |
| CDPG | 110.55 | 73.69 | 7.34 | 19.96 | 15.47 | 35.34 | 18.23 | 8.78 | 5.38 |
| – AcyM | 120.52 | 80.03 | 8.88 | 22.32 | 17.49 | 38.61 | 20.13 | 10.13 | 6.29 |
| – URP | 181.94 | 88.44 | 7.61 | 20.08 | 18.31 | 44.79 | 20.98 | 8.76 | 5.51 |
| – BNE | 105.47 | 70.65 | 7.36 | 19.48 | 15.19 | 34.19 | 17.76 | 8.69 | 5.36 |

$d = \log_2 n$. Rank of the node embedding matrix $l = 8$, learning rate $\eta = 0.06$. We use the RMSprop optimizer for training. All experiments were conducted using the GPU: NVIDIA Tesla V100-SXM2.

### 4.2 OVERALL PERFORMANCE

Table 1 reports the effectiveness of our method against baselines in terms of the objective value $J$, averaged over 20 runs to account for stochasticity. Lower $J$ indicates better performance. Table 2 presents the corresponding average runtime. Across all instances, CDPG consistently achieves the lowest $J$, improving over the best baseline (denoted with $^\dagger$) by 2.50%–27.67%. The variation in improvement is largely explained by the behavior of the Held algorithm, whose approximation ratio is bounded by $\mathcal{O}(\log |S|)$. On small graphs (e.g., Graphs 3, 8, and 9), Held already produces solutions close to the optimum, leaving limited room for improvement. In contrast, as graph size increases, its approximation bound becomes looser, and CDPG demonstrates substantially larger advantages, underscoring its scalability to large-scale networks.

On large graphs, gradient-based methods achieve better performance; however, a substantial gap remains compared to CDPG, which further improves with increasing $n$. Moreover, the runtime of MEP and GS is close to that of CDPG, yet the results confirm that CDPG's decision paradigm is more effective, achieving better solutions with consistent convergence speed. As for Notears, although it performs well on several graphs, its runtime is prohibitively large and training is slow, highlighting the advantage of CDPG in employing an acyclic mask to enforce cycle constraints.

**Evaluation under Varying Source Set Sizes:** To further assess the performance disparity between Held and CDPG across different scales of source sets, we conducted additional experiments by setting part of the sources as non-sources for fair comparison on five datasets. Figures 2 and 3 systematically illustrate the results, showing that the loss of CDPG increases more slowly with $|S|$, while its runtime remains nearly constant. In contrast, the runtime of the Held method grows linearly

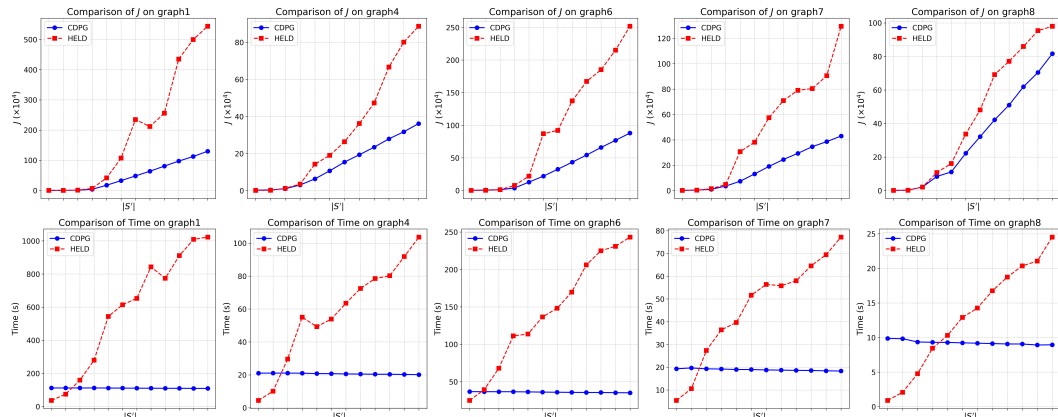

Figure 2: Comparison of time cost and metric $J$ between Held and CDPG across five graphs. The x-axis ($|S'|$) shows the count of randomly sampled source nodes, with values (left to right): $0.5 \log_2 n$, $\log_2 n$, $4 \log_2 n$, $16 \log_2 n$, $0.1|S|$, $0.2|S|$, ..., $0.8|S|$.

with $|S|$. Although Held performs faster at smaller subset sizes, CDPG begins to outperform it once the subset size exceeds a critical threshold—on the order of a small constant multiple of $\log_2 n$.

Overall, the results demonstrate that CDPG surpasses existing optimization methods in both solution quality and computational efficiency, and its balanced performance establishes it as a strong candidate for solving Cost-Distance problems.

### 4.3 PARAMETER ANALYSIS

As shown in Table 5, we study the effect of rank of node embedding matrix $l \in \{2, 4, 6, 8, 10, 12\}$ and learning rate $\eta \in \{0.02, 0.06, 0.10, 0.14\}$ in graph 3. CDPG shows stable performance across most settings. The best objective value (24.63) is achieved when $\eta = 0.06$ and $l = 8$. Based on this, we adopt this configuration as the default in all experiments.

We compare several common optimizers, including Adagrad Duchi et al. (2011), Adam Kingma (2014), NAdam Dozat (2016), and RMSprop Tieleman (2012), as shown in Figure 4. RMSprop achieves the lowest final loss and converges the fastest among all candidates in graph 3. Based on this empirical observation, we adopt RMSprop as the default optimizer.

We study the impact of the loss iteration depth $d$ in graph 6, normalized by $\log_2 n$, on both runtime and loss (Figure 5). As $d$ increases, loss decreases rapidly and stabilizes near $d/\log_2 n = 1$, while runtime grows linearly. Setting $d = \log_2 n$ achieves a favorable trade-off between performance and efficiency, and is therefore adopted as the default configuration.

### 4.4 ABLATION STUDIES

We conduct a comprehensive ablation study by removing or replacing specific modules. The results in terms of objective value $J$ are in Table 1, and runtimes are in Table 2. Three ablated variants are considered. **–AcyM:** Ablation of the acyclic mask $A$. **–URP:** Ablation of the unreached penalty $\mu$. **–BNE:** Ablation of the bilinear node embedding $Q, K$, only preserves edge logits $H$.

The full CDPG model consistently achieves the best $J$ values across all instances, confirming the necessity of each component. –AcyM results in a notably higher $J$, and also slightly increases the runtime due to convergence issues. –URP results in similar performance to the full model, but a clear rise in runtime, confirming its role in accelerating optimization without significantly harming performance. –BNE slightly increases the objective value, especially in large graphs, indicating that BNE helps escape poor local minima in optimization.

In summary, each module contributes uniquely to CDPG's performance. The synergy of the acyclicity constraint, fast-converging URP, and direction-aware embeddings is key to the method's effectiveness and efficiency.

## 5 CONCLUSION

We proposed CDPG, the first gradient-based framework for the Cost-Distance problem, which enables efficient and differentiable optimization via a policy gradient approach. By introducing a novel acyclic mask, unreached penalty, and bilinear node embedding, CDPG not only converges time complexity of $\mathcal{O}(m \log n)$, but also achieves strong performance on real-world UAV logistics networks. Looking forward, we plan to extend CDPG to unsupervised training of a generalizable model without relying on combinatorial search for labels.

## 6 REPRODUCIBILITY STATEMENT

To ensure reproducibility, we provide detailed descriptions of the baseline implementations in Appendix E and comprehensive dataset information in Appendix D. The full source code of our method is available at `https://anonymous.4open.science/r/iclr_cdpg-8737`. Due to the restrictions of anonymous GitHub, only a subset of the datasets can be hosted at this time; however, upon acceptance and publication, we will make all datasets publicly available to ensure full reproducibility.

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

## A RELATED WORK

We provide the full version of the related work here, while the most important parts have already been presented in the introduction.

### A.1 MODEL

**Hub Location and Routing Problem(HLRP):** The HLRP integrates hub location optimization with multi-echelon routing planning, addressing challenges ranging from infrastructure cost minimization in the p-Hub Median Problem O'kelly (1986); Wu et al. (2022) to service equity optimization in the p-Hub Center Problem Kartal et al. (2019). Recent advancements have extended these models to low-carbon transportation systems Gao et al. (2022) and adaptive network design for dynamic demand Wu et al. (2024), leveraging metaheuristics like genetic algorithms Bostel et al. (2015) and tabu search Bütün et al. (2021).

**Vehicle Routing Problem(VRP):** The VRP, since its seminal formulation by Dantzig and Ramser Dantzig & Ramser (1959), has evolved to address rich variants including collaborative multi-depot systems Wang et al. (2020) and open-route configurations Repoussis et al. (2010), with hybrid metaheuristics like firefly algorithms Altabeeb et al. (2019) enhancing solution robustness.

**Other models about UAV network:** Existing UAV logistics studies can be broadly categorized into air–ground collaborative routing Feng & Cao (2025); Rashid Nazir et al. (2025); Qi et al. (2025); Li et al. (2022), dynamic delivery under uncertainty such as weather, road fragility, and risk costs Mahmoodi et al. (2025); Shao et al. (2022), and traffic- or congestion-aware routing frameworks Zhang et al. (2015); She & Ouyang (2021); despite heterogeneity in application scenarios and operational constraints, all these works account for both route planning costs and UAV flow transportation costs, which are abstracted by the Cost-Distance problem.

### A.2 METHOD

**Solving Cost-Distance Problem:** There has been significant progress on approximation algorithms for this problem. Meyerson et al. (2008) proposed the first randomized hierarchical aggregation algorithm with an $\mathcal{O}(\log |S|)$-approximation ratio and a running time of $\mathcal{O}(|S|^2(m + n \log n))$. Chekuri et al. (2001) later provided a derandomized version using a linear-programming dual construction, making the algorithm more controllable. More recently, Held & Perner (2025) showed that the running time can be improved to $\mathcal{O}(|S|(m + n \log n))$. On the hardness side, Chuzhoy et al. (2008) proved that this problem cannot be approximated better than $\Omega(\log \log n)$ unless $\text{NP} \subseteq \text{DTIME}(n^{\mathcal{O}(\log \log \log n)})$.

**Machine learning for combinatorial optimization (ML4CO).** Recent years have seen a surge of ML-based methods for combinatorial optimization problems (ML4CO). Two broad paradigms emerged: (i) learning heuristics/constructive policies that build solutions incrementally using graph embeddings and reinforcement learning (or imitation) Khalil et al. (2017); Kool et al. (2018), and (ii) learning to augment exact solvers (e.g., learning branching rules for branch-and-bound) to accelerate classical algorithms Gasse et al. (2019); Gupta et al. (2020). Representative works include Khalil et al. (2017). Who learn greedy construction policies with graph embeddings and RL, Kool et al.

(2018) attention-based pointer networks trained with REINFORCE for routing problems, and a line of work learning branching/variable-selection policies for MIP solvers Gasse et al. (2019); Gupta et al. (2020). More recently, methods that combine learning with classical heuristics/local-search to scale to very large VRPs (e.g., learning to select subproblems to "delegate") have been proposed Li et al. (2021). These approaches are powerful when (a) problem instances share a distribution so learned policies generalize, or (b) fast approximate solutions (or solver speedups) are acceptable.

**Continuous relaxations and differentiable combinatorial layers.** A second research thrust formulates combinatorial constraints/objects as differentiable surrogates so gradients can be used directly. Examples include continuous acyclicity penalties for DAG learning Zheng et al. (2018); Yu et al. (2019; 2021), differentiable sampling/relaxations such as Gumbel-Softmax Jang et al. (2016), maximum-entropy policy parametrizations Haarnoja et al. (2018), and differentiable combinatorial operators (e.g., differentiable minimum-spanning-forest / perturbed spanning forests) that permit end-to-end gradient flow Stewart et al. (2023). Another related line uses path-based relaxations and gradient-projection schemes to handle routing / system-optimum problems with heterogeneous user valuations Wu & Huang (2014). Recent work also studies learned differentiable shortest-path approximations Lahoud et al. (2024).

# B  PROOFS

## B.1  PROOF OF THEOREM 1

*Proof.* Let $\mathcal{P}_s$ denote the shortest path from source node $s$ to the root node $r$ under the edge cost matrix $\Phi^*$. Assume that the optimal solution to the Cost-Distance problem routes the entire source weight $w_s$ along $\mathcal{P}_s$. Then, for each edge $(i,j) \in \mathcal{P}_s$, the flow on that edge increases by $w_s$, and the total edge flow becomes:

$$W_{ij}^* = \sum_{s:(i,j)\in\mathcal{P}_s} w_s.$$

Now consider the Cost-Distance objective:

$$J(X') = \sum_{(i,j)} X'_{ij}(C_{ij} + f_{ij}D_{ij}),$$

where $X' \in \{0,1\}^{n\times n}$ indicates which edges are selected, and $f_{ij}$ is the total weight that traverses edge $(i,j)$.

If all source weights $w_s$ are routed along their respective shortest paths $\mathcal{P}_s$, then the edge flows $f_{ij}$ equal $W_{ij}^*$, and the total cost becomes:

$$J(W^*) = \sum_{(i,j)} W_{ij}^* D_{ij} + \sum_{(i,j)} C_{ij}.$$

Next, define a new edge cost matrix:

$$\Phi_{ij}^* = \frac{C_{ij}}{W_{ij}^* + \varepsilon} + D_{ij},$$

where division is element-wise. Then for each source node $s$, its shortest path cost under $\Phi^*$ is:

$$\sum_{(i,j)\in\mathcal{P}_s} \Phi_{ij}^* = \sum_{(i,j)\in\mathcal{P}_s} \left(\frac{C_{ij}}{W_{ij}^* + \varepsilon} + D_{ij}\right).$$

Multiplying this path cost by the source weight $w_s$, and summing over all $s$, we obtain the total cost:

$$\sum_s w_s \sum_{(i,j)\in\mathcal{P}_s} \left(\frac{C_{ij}}{W_{ij}^* + \varepsilon} + D_{ij}\right) = \sum_{(i,j)} \left(\frac{C_{ij}}{W_{ij}^* + \varepsilon} \sum_{s:(i,j)\in\mathcal{P}_s} w_s + W_{ij}^* D_{ij}\right)$$

$$= \sum_{(i,j)} \left(\frac{C_{ij}W_{ij}}{W_{ij}^* + \varepsilon} + W_{ij}^* D_{ij}\right)$$

$$\approx J(W^*)$$

Thus, minimizing the Cost-Distance objective is equivalent to minimizing the total weighted path cost under $\Phi^*$, assuming the weights are routed along shortest paths. Therefore, using Dijkstra's algorithm under $\Phi^*$ to obtain the paths $\{\mathcal{P}_s\}$ recovers the same edge selections as the optimal solution $X'$. Thus, we have

$$W^* = \text{Dijkstra}(\Phi^*, r), \quad \text{where } \Phi^* := W^{*\odot -1} \odot C + D.$$

$\square$

### B.2 Proof of Theorem 2

*Proof.*

**Lemma 1** (Bellman equation). *The value function $V_\gamma(s)$ for a given policy $\pi$ satisfies the Bellman equation Bellman (1966):*

$$V_\gamma(s) = \mathbb{E}_{a \sim \pi(\cdot|s)} \left[ r(s) + \gamma \sum_{s' \in S} p(s'|s,a) V_\gamma(s') \right], \tag{18}$$

In our MDP setup, the state to which the agent transitions after taking an action is deterministic, based on the policy, thus we have $P_{ss'}^\theta = p(s'|s,a)$. Therefore, according to Lemma 1, the vector form of the Bellman equation is

$$\mathbf{V}_\gamma(\mathbf{s}) = \mathbb{E}[\mathbf{r}] + \gamma P^\theta \mathbf{V}_\gamma(\mathbf{s}), \tag{19}$$

where $\mathbf{r}$ is the vector form of $r(s)$. Then,

$$\mathbf{V}_\gamma(\mathbf{s}) = -(P^\theta \odot F) \cdot \mathbf{1}_n + \gamma P^\theta \mathbf{V}_\gamma(\mathbf{s}),$$
$$\Rightarrow (I - \gamma P^\theta)\mathbf{V}_\gamma(\mathbf{s}) = -(P^\theta \odot F) \cdot \mathbf{1}_n,$$
$$\Rightarrow \mathbf{V}_\gamma(\mathbf{s}) = -(I - \gamma P^\theta)^{-1}(P^\theta \odot F) \cdot \mathbf{1}_n. \tag{20}$$

**Note:** $I - \gamma P^\theta$ is invertible when $\gamma = 1$.

Since $P^\theta$ is a directed acyclic graph, its eigenvalues $\lambda$ satisfy $\lambda = 0$ which will be proved in Appendix A1.2. The eigenvalues of $I - \gamma P^\theta$ are $1 - \gamma\lambda$, which are strictly positive for all $\gamma = 1$. Since the eigenvalues of $I - P^\theta$ are non-zero, the matrix $I - P^\theta$ is full rank and invertible.

$\square$

### B.3 Proof of Theorem 3

**Lemma 2.** *A graph that can be topologically sorted is acyclic.*

**Lemma 3** (Cayley-Hamilton Theorem). *Let $A$ be an $n \times n$ matrix with characteristic polynomial*

$$p(\lambda) = \det(A - \lambda I),$$

*where $I$ is the identity matrix and $\lambda$ is is an arbitrary eigenvalue of $A$. The Cayley-Hamilton theorem asserts that $A$ satisfies its own characteristic equation:*

$$p(A) = 0,$$

**Lemma 4** (Nilpotent matrix property). *Let $A$ be an $n \times n$ matrix. The matrix $A$ is a nilpotent matrix if and only if there exists $k$ such that $A^k = 0, k > 0$. Moreover, a matrix $A$ is nilpotent if and only if all of its eigenvalues are zero.*

*Proof.* To prove Equation (10), we only have to prove that

$$P^{\theta^t} = 0, \quad \forall t \geq |\mathcal{V}|, \tag{21}$$

which indicates that only $|\mathcal{V}|$ iterations are needed in Nenmann iteration.

Since Equation (5), all the stations in the graph have been ordered based on their distance from the central station, with only edges pointing from farther stations to closer stations being retained. This distance-based ordering forms a topological sorting sequence, ensuring that the graph is acyclic according to Lemma 2, which means

$$\text{tr}(P^{\theta\,t}) = 0, \quad \forall t > 0,$$

$$\Rightarrow \sum_{i=1}^{\text{rank}^t} \lambda_i^t = 0, \quad \forall t > 0,$$

where $\text{rank}^t$ denotes the number of eigenvalue of $P^{\theta\,t}$. We set $t = 2$ and get

$$\sum_{i=1}^{\text{rank}^2} \lambda_i^2 = 0$$

$$\Rightarrow \forall i \in [1, \text{rank}^2], \quad \lambda_i^2 = 0.$$

Therefore, $P^{\theta\,2}$ is a nilpotent matrix and because of Lemma 4, there exists $k > 0$, $P^{\theta\,2k} = 0$. Thus $P^\theta$ is a nilpotent matrix too.

Accroding to Lemma 3, we have

$$p(\lambda) = \det(P^\theta - \lambda I) = \lambda^{|\mathcal{V}|} = 0$$

$$\Rightarrow p(P^\theta) = P^{\theta|\mathcal{V}|} = 0$$

$$\Rightarrow P^{\theta\,t} = 0, \quad \forall t \geq |\mathcal{V}|$$

$\square$

### B.4 Proof of Theorem 4

*Proof.* The proof is structured in three main steps. First, we establish the Lipschitz continuity of the gradient with respect to the probabilistic matrix $P$, denoted $\nabla_P \mathbb{E}[J(P)]$. Second, we show that the intermediate operations (masking and softmax) are 1-Lipschitz. Third, we translate the Lipschitz constant from the space of $P$ to the parameter space of $\theta = \{H, Q, K\}$.

#### Step 1: Lipschitz Continuity of $\nabla_P J(P)$

The objective function $J(P)$ is decomposed as:

$$J(P) = \Omega_C(P) + \Omega_D^{\text{trunc}}(P) + \mu(P)$$

where:

$$\Omega_C(P) = \text{tr}(P^T C), \quad \Omega_D^{\text{trunc}}(P) = -w^\top \left( \sum_{t=1}^{d} P^t \odot D \right) \mathbf{1}_n, \quad \mu(P) = w^T P^d L_D^X.$$

We analyze the gradient of each component separately.

#### Analysis of $\Omega_C(P)$

Since $\Omega_C(P)$ is a linear function of $P$, its gradient is the constant matrix $C$. Therefore, its contribution to the Lipschitz constant is zero:

$$\|\nabla_P \Omega_C(P_1) - \nabla_P \Omega_C(P_2)\|_F = \|C - C\|_F = 0.$$

#### Analysis of $\mu(P)$

The gradient of $\mu(P)$ involves the matrix power $P^d$. Using the matrix power difference inequality, we can bound the change in the gradient:

$$\|\nabla_P \mu(P_1) - \nabla_P \mu(P_2)\|_F \leq d^2 \rho^{d-1} \|w\| \|L_D^X\| \|P_1 - P_2\|_F.$$

ANALYSIS OF $\Omega_D^{\text{TRUNC}}(P)$

The gradient of this term is $\nabla_P \Omega_D^{\text{trunc}} = -\sum_{t=1}^d \nabla_P(w^T(P^t \odot D)\mathbf{1}_n)$. By analyzing the derivative of matrix powers, we derive the following bound:

$$\|\nabla_P \Omega_D^{\text{trunc}}(P_1) - \nabla_P \Omega_D^{\text{trunc}}(P_2)\|_F \leq \|w\|\|D\|_F \left(\sum_{t=1}^d t^2 \rho^{t-1}\right)\|P_1 - P_2\|_F.$$

JUSTIFICATION FOR $\rho < 1$

**Lemma 5.** *Let $P$ be the transition matrix of a directed acyclic graph (DAG) where the root node is a sink (no outgoing edges). Then, the maximum Frobenius contraction rate $\rho < 1$.*

*Proof.* Assume, for the sake of contradiction, that $\rho \geq 1$. This implies the existence of a $t \geq 1$ such that $\|P^{t+1}\|_F \geq \|P^t\|_F$. Let $v_i^{(t)}$ be the $i$-th row of $P^t$. Then $\|P^t\|_F^2 = \sum_i \|v_i^{(t)}\|_2^2$. The update rule is $v_i^{(t+1)} = v_i^{(t)} P$. Since the root is a sink, its corresponding row in $P$ is zero. For any non-root node $i$, the update $v_i^{(t+1)}$ is a convex combination of the rows of $P$, which includes the zero row with a strictly positive weight. This leads to a strict contraction, $\|v_i^{(t+1)}\|_2 < \|v_i^{(t)}\|_2$ for all non-root $i$. For the root node, the norm remains zero. Summing over all rows yields $\|P^{t+1}\|_F^2 < \|P^t\|_F^2$, which contradicts our initial assumption. Thus, $\rho < 1$. $\qquad\square$

AGGREGATED LIPSCHITZ BOUND FOR $\nabla_P J(P)$

Combining the bounds for all components, we get the Lipschitz constant for $\nabla_P J(P)$, denoted as $L_P$:

$$L_P \leq \|w\|\|D\|_F \sum_{t=1}^d t^2 \rho^{t-1} + \|w\|\|L_D^X\|d^2\rho^{d-1}.$$

Since $\rho < 1$, we can bound the series and the term dependent on $d$ with constants independent of $d$:

$$\sum_{t=1}^\infty t^2 \rho^{t-1} = \frac{1+\rho}{(1-\rho)^3}, \quad \sup_{d \geq 1} d^2 \rho^{d-1} = \frac{4}{(1-\rho)^2 e}.$$

This provides a global upper bound for $L_P$.

STEP 2: IMPACT OF MASKING AND SOFTMAX OPERATIONS

The transformation from the parameterized matrix $P^\theta$ to the final probability matrix $P$ involves masking and a row-wise softmax. Both operations are 1-Lipschitz.

- **Masking**: As a linear projection, $\|(P_1 - P_2) \odot A \odot X\|_F \leq \|P_1 - P_2\|_F$.

- **Row-wise Softmax**: The Jacobian of the softmax function, $\text{Jacobian}(z) = \text{diag}(s) - ss^\top$ where $s = \text{EdgeSoftmax}(z)$, has a spectral norm of at most 1, making the function non-expansive. Thus, $\|\text{EdgeSoftmax}(P_1) - \text{EdgeSoftmax}(P_2)\|_F \leq \|P_1 - P_2\|_F$.

Therefore, these steps do not increase the Lipschitz constant.

STEP 3: TRANSLATION FROM $L_P$ TO $L$

The parameterization is $P^\theta = H + l^{-1/2}Q^T K$. The norm of the full gradient $\nabla_\theta J$ is related to the norm of $\nabla_{P^\theta} J$ by:

$$\|\nabla_\theta J\|_F^2 = \|\nabla_H J\|_F^2 + \|\nabla_Q J\|_F^2 + \|\nabla_K J\|_F^2 \leq (1 + 2l^{-1/2})\|\nabla_{P^\theta} J\|_F^2$$

where we have used the normalization condition $\|Q\|_2 = \|K\|_2 = 1$. This implies that the Lipschitz constant for $\nabla_\theta J$ is scaled by a factor of $\sqrt{1 + 2l^{-1/2}}$. Combining this with the bound on $L_P$ completes the proof. $\qquad\square$

## B.5 PROOF OF THEOREM 5

Let $\hat{L}_D^{X\odot A}$ be the longest distance of nodes $\mathcal{V} - \{r\}$ to root node $r$. First, we have,

$$w^\top L_D^{X\odot A}(\mathcal{V}, r) \leq \mathbb{E}[\Omega_D] = w^\top \sum_{t=1}^n P^{\theta t} \odot D\mathbf{1}_n \leq w^\top \sum_{t=1}^d P^{\theta t} \odot D\mathbf{1}_n + w^\top P^{\theta d} \hat{L}_D^{X\odot A}(\mathcal{V}, r)$$

Then, the bias satisfies

$$\frac{\mathbb{E}[\Omega_D] - \mathbb{E}[\Omega_D^{\text{trunc}}]}{\mathbb{E}[\Omega_D]} = \frac{w^\top \sum_{t=1}^n P^{\theta t} \odot D\mathbf{1}_n - w^\top \sum_{t=1}^d P^{\theta t} \odot D\mathbf{1}_n - w^\top P^{\theta d} L_D^{X\odot A}(\mathcal{V}, r)}{w^\top \sum_{t=1}^n P^{\theta t} \odot D\mathbf{1}_n},$$

$$= \frac{w^\top \sum_{t=d+1}^n P^{\theta t} \odot D\mathbf{1}_n - w^\top P^{\theta d} L_D^{X\odot A}(\mathcal{V}, r)}{w^\top \sum_{t=1}^n P^{\theta t} \odot D\mathbf{1}_n},$$

$$\leq \frac{w^\top P^{\theta d} \hat{L}_D^{X\odot A}(\mathcal{V}, r) - w^\top P^{\theta d} L_D^{X\odot A}(\mathcal{V}, r)}{w^\top L_D^{X\odot A}(\mathcal{V}, r)}$$

$$\leq \frac{w^\top P^{\theta d} \hat{L}_D^{X\odot A}(\mathcal{V}, r)}{w^\top L_D^{X\odot A}(\mathcal{V}, r)}$$

$$\leq \frac{\left(\frac{w^\top}{\|w\|}\right) P^{\theta d} (n\|D\|_{\max}\mathbf{1}_n)}{\left(\frac{w^\top}{\|w\|}\right) (\|D\|_{\min}\mathbf{1}_n)}$$

$$\leq n\frac{\|D\|_{\max}}{\|D\|_{\min}}\rho^d \leq \epsilon$$

Then, if we want to bound bias within $\epsilon$, $d$ satisfies

$$d \geq \log_{\frac{1}{\rho}}\left(n\frac{\|D\|_{\max}}{\|D\|_{\min}}\right) \propto \log n.$$

## B.6 PROOF OF THEOREM 6

We analyze the convergence of vanilla gradient descent for the objective function $J(\theta)$, assuming the gradient is $L$-Lipschitz continuous as established in Theorem 4. For notational simplicity, we use $J(\theta)$ to replace $\mathbb{E}[J(\theta)]$. For all $P_1, P_2 \in \mathbb{R}^{n\times n}$, we assume:

$$\|\nabla J(\theta_1) - \nabla J(\theta_2)\|_F \leq L\|\theta_1 - \theta_2\|_F.$$

We consider the standard gradient descent update rule:

$$\theta_{t+1} = \theta_t - \eta\nabla J_{\theta_t},$$

where $\eta \in (0, 1/L]$ is the step size. The following classical inequality holds for smooth functions with $L$-Lipschitz gradient (see e.g. (Bottou, Curtis, and Nocedal 2018)):

$$J(\theta_{t+1}) \leq J(\theta_t) + \langle\nabla J_{\theta_t}, \theta_{t+1} - \theta_t\rangle + \frac{L}{2}\|\theta_{t+1} - \theta_t\|_F^2$$

$$= J(\theta_t) - \eta\|\nabla J_{\theta_t}\|_F^2 + \frac{L}{2}\eta^2\|\nabla J_{\theta_t}\|_F^2$$

$$= J(\theta_t) - \eta\left(1 - \frac{L\eta}{2}\right)\|\nabla J_{\theta_t}\|_F^2.$$

When $\eta \leq \frac{1}{L}$, the quantity in parentheses is nonnegative, so this ensures that $J(\theta_t)$ decreases monotonically.

Let $\theta^* = \theta_{\text{loc}}$ be a stationary point such that $\nabla J(\theta^*) = 0$. Then, by applying a classical result in smooth convex analysis (even if $J$ is non-convex, this still gives a rate in terms of stationarity or gap to the stationary value), we have:

$$J(\theta_T) - J(\theta^*) \leq \frac{L}{2T}\|\theta_0 - \theta^*\|_F^2. \tag{22}$$

Note that the optimization goal is to find a policy matrix $\theta$ that minimizes the objective $J(\theta)$, but not necessarily to evaluate $J$ itself.

If we scale $J$ by a constant positive factor $\alpha > 0$, i.e., let $\tilde{J} = \alpha J$, then the gradient becomes $\nabla \tilde{J} = \alpha \nabla J$, and the update rule becomes:

$$\theta_{t+1} = \theta_t - \eta \nabla_{\theta_t} \tilde{J} = \theta_t - \eta \alpha \nabla_{\theta_t} J.$$

This is equivalent to performing gradient descent on $J$ with effective step size $\tilde{\eta} = \alpha \eta$, so the iteration trajectory is preserved modulo step size rescaling.

Moreover, in adaptive optimizers such as Adam, even this step-size difference is absorbed through internal rescaling, so the optimizer's behavior is invariant to scalar scaling of the objective. Therefore, in our analysis and design, we may ignore constant multiplicative factors in $L$, and report iteration complexity as:

$$T = \mathcal{O}(1).$$

$\square$

### B.7 Proof of Theorem 7

To theoretically analyze the fastest possible time lower bound, we assume that all involved matrices are stored in an adjacency list (sparse) format. This enables:

**Sparse Matrix Assumption.**

- Matrix addition in $O(m)$,
- Matrix-vector or matrix-matrix multiplication in $O(m)$,

where $m$ is the number of non-zero entries (edges) in the graph. In practice, the multiplication between matrices and vectors is carried out on the GPU. Owing to its high degree of parallelism, the actual running time is significantly smaller than that of operations on sparse graphs.

**Acyclic Mask Computation.** CDPG requires generating an acyclic transition matrix $P^\theta$, which relies on constructing a shortest-path tree rooted at a central station. This is implemented via Dijkstra's algorithm with a Fibonacci heap, giving a complexity of:

$$O(m + n \log n).$$

**Forward Propagation.** According to Equation (11) and Theorem 3, computing the truncated value term $\sum_{t=1}^{d} P^t \odot D \mathbf{1}_n$ involves $d$ matrix-vector products. Each multiplication $P^t \cdot v$ (with vector or dense matrix) requires $O(m)$, leading to:

$$O(d\,m).$$

In addition, the penalty term $\mu^\theta = w^\top P^d L_D^X$ involves one sparse matrix-vector product $O(m)$. The total forward pass is: $O(d\,m)$.

**Backward Propagation.** The gradient of $J(P^\theta)$ is computed through automatic differentiation (backpropagation). Since all operations in the forward pass are differentiable and structurally mirrored in reverse (matrix-vector products and element-wise operations), the backward pass has the same complexity:

$$O(d\,m).$$

**Total Over $T$ Iterations.** Performing forward and backward passes over $T$ optimization steps gives:

$$T \cdot (O(d\,m) + O(d\,m)) = O(T\,d\,m).$$

**Total Complexity.** Combining all components, the total computational complexity of CDPG is:

$$O(T\,d\,m) + O(m + n \log n) = O(m \log n),$$

assuming a constant number of optimization steps $T = O(1)$ (as justified in Theorem 6), and $d = O(\log n)$.

---

**Algorithm 1** Routing Distance Estimation

---

**Input**:policy matrix $P$, distance iteration $d$, the shortest distance to root $L_D^{X \odot A}(\mathcal{V}, r)$
**Output**: $\mathbb{E}[\Omega_D^{trunc}]$
1: Initialize $\mathbb{E}[\Omega_D^{\text{trunc}}] \leftarrow 0$, $w' \leftarrow w$, $k \leftarrow (P \odot D) \cdot \mathbf{1}_n$, $i \leftarrow 1$
2: **while** $i \leq d$ **do**
3:    $\mathbb{E}[\Omega_D^{trunc}] \leftarrow \mathbb{E}[\Omega_D^{trunc}] + w'^\top k$
4:    $w' \leftarrow Pw'$   # parallelized $\mathcal{O}(m)$ operations.
5:    $i \leftarrow i + 1$
6: **end while**
7: $\Omega_D^{trunc} \leftarrow \Omega_D^{trunc} + w'^\top L_D^{X \odot A}(\mathcal{V}, r)$
8: **return** $\Omega_D^{trunc}$

---

Table 3: Statistics of Each graph

| Statistic | Graph 1 | Graph 2 | Graph 3 | Graph 4 | Graph 5 | Graph 6 | Graph 7 | Graph 8 | Graph 9 |
|---|---|---|---|---|---|---|---|---|---|
| $n$ | 17408 | 13687 | 2650 | 6145 | 5138 | 8834 | 5737 | 3126 | 1629 |
| $m$ | 1137357 | 733838 | 84694 | 237055 | 172451 | 399774 | 185772 | 88638 | 41043 |
| $|S|$ | 16440 | 12798 | 2346 | 5473 | 4489 | 8123 | 5204 | 2691 | 1174 |

**Parallelism.** All matrix-vector and matrix-elementwise operations are highly parallelizable on modern GPU hardware. Thus, CDPG achieves both theoretical efficiency and practical scalability.

$\square$

## C   Overall Learning Process of CDPG

The overall learning process of CDPG is shown in Algorithm 2.

## D   Details of the Real World Graph Dataset

### D.1   Scale of datasets

The scale of our dataset is shown in Table 3.

### D.2   Spatial Sampling and Core Node Selection

The locations with logistics attributes in the Guangdong–Hong Kong–Macao Greater Bay Area are traversed and searched with an accuracy of 0.25 square kilometers. Only one point is retained within every 2500 square meters. Eight nodes are artificially selected as the first-level nodes.

Table 4: Performance of DJI FlyCart 30

| Item | Value | Unit |
|---|---|---|
| Battery energy | 7200000 | J |
| Maximum range | 28 | KM |
| Flight speed | 20 | m/s |
| Battery cycle times | 1500 | times |
| Battery price | 10000 | yuan |
| Take-off energy consumption | 11200 | J |
| Flight cost per kilometer | 0.085 | yuan |

---

**Algorithm 2** CDPG

---

**Input**: adjacency matrix $X$, cost weight $C$, distance weight $D$, node weight $w$
**Output**: $X'$

1: Initialize policy parameters $\theta = \{H, Q, K\}$:
   $H, Q \sim \mathcal{N}(0, 1), K \leftarrow 0$, learning rate $\eta \leftarrow 0.06$, distance iteration $d \leftarrow \log_2 n$, node embedding rank $l \leftarrow 8$
2: Compute the shortest distance to root using weight $\Phi$:
   $\Phi = \frac{\|w\|_1}{2n} C + D$
   $L_\Phi^X(\mathcal{V}, r) \leftarrow \text{Dijkstra}(\Phi, X)$
3: Generate $A$ from $L_\Phi^X(\mathcal{V}, r)$ using the formula (5)
4: Compute the shortest distance to root on an acyclic graph: $L_D^{X \odot A}(\mathcal{V}, r) \leftarrow \text{Dijkstra}(D, X \odot A)$
5: **for** $i \leftarrow 1$ **to** $T$ **do**
6:    Generate policy matrix $P$ from $\theta$:
      $P \leftarrow H + l^{-1/2} Q^\top K$
      $P \leftarrow P \odot X \odot A$
7:    Normalize $P$ with edge-softmax from formula (6)
8:    Compute construction cost:
      $\Omega_C \leftarrow \text{tr}\left(P^\top C\right)$
9:    Compute routing distance using algorithm 1:
      $\Omega_D \leftarrow \text{Routing Distance Estimation}(P, d, L_D^{X \odot A}(\mathcal{V}, r))$
10:   $J \leftarrow \Omega_C + \Omega_D$
11:   Update policy $\theta$:
      $\theta \leftarrow \theta - \alpha \frac{\partial J(\theta)}{\partial \theta}$
12: **end for**
13: Generate $P$ from $\theta$:
    $P \leftarrow H + \frac{1}{\sqrt{l}} Q^\top K$
    $P \leftarrow P \odot X \odot A$
14: Generate $X'$ from $P$ using edge-argmax from (7)
15: Pruning policy matrix $X'$:
16: **while** $\exists x \in \mathcal{V} : w'_x = 0$ and $\sum_{y \in \mathcal{V}} X'_{y,x} = 0$ **do**
17:   $X'_{x,:} \leftarrow 0$
18: **end while**
19: **return** $X'$

---

### D.3 NODE WEIGHT

Mobile signaling data is generally used in the field of transportation to determine the demand between nodes Li et al. (2024) . The simulation utilizes mobile signaling data for online shopping activities, obtained from China Unicom Smart Footprint Data Technology Co., Ltd. (August 2022 dataset). To accommodate the block-structured nature of the data, we assign each network node a standardized service area of 1 km². In high-density urban zones where multiple nodes fall within a 1000 m² region, the total service demand is equally distributed among all nodes located in that area.

### D.4 DISTANCE WEIGHT AND COST WEIGHT

In highly urbanized areas such as Singapore, UAV flight paths are usually planned above existing urban roads Tan et al. (2019). Therefore, based on the high urbanization rate of the Greater Bay Area, the walking distance between nodes is taken as the transportation distance. Considering the high complexity of obtaining the global distance matrix, this study first crawls the distance between each primary node and all other nodes. For the remaining edges, this study crawls them in the order of straight-line distances of 500 meters, 1000 meters, and 1500 meters. Finally, the walking distances between all nodes within a straight-line distance of 2000 meters are collected. The cost associated with this problem comprises two components: transportation cost and construction cost, which is 0.1 in this Dataset.

The construction cost is proportional to the path length. It depends on the construction cost per unit length, which remains constant for each path.

Drone transportation is divided into trunk transportation and feeder transportation, and their transportation costs are different Liu (2021) . The transportation cost of feeder drones is usually composed of battery loss and electricity cost Dukkanci et al. (2024) . This study selects DJI FlyCart 30 as an example, and its specific performance is as shown in Table 4.

This study selects "Fengniao H-VTOL" as the unmanned aerial trunk transport vehicle, and its transportation cost is three to four times that of ordinary truck transportation Liu (2021) . Based on the drone performance mentioned in the previous text, we construct a transportation cost function:

$$C_{tot} = \begin{cases} (C_{fly} \times D) + \left( \lceil \frac{E_{TO} + E_{/km} \times D}{E_{tot}} \rceil \times \frac{C_{batt}}{N_{cyc}} \right), & D \leq 28 \\ (4 - \frac{D-28}{300-28}) \times C_{trk} \times D, & D > 28 \end{cases}$$

where $C_{fly}$ represents cost per kilometer of flight, $E_{TO}$ represents takeoff energy consumption, $E_{/km}$ represents power consumption per kilometer, $E_{tot}$ represents battery total energy, $C_{batt}$ represents battery price, $N_{cyc}$ represents battery cycle life, $C_{trk}$ represents transportation cost per kilometer of the truck.

# E    DETAILS OF OTHER METHODS

To guarantee the reproducibility and fair evaluation of all experimental results, we provide a detailed overview of the parameter settings used for all methods employed in this study. Each method was configured using either default values commonly reported in the literature or optimized through grid search or empirical validation on the validation set.

## E.1    HEURISTIC SEARCH

- **Genetic Algorithm (GA)** This method applies a Genetic Algorithm to optimize graph structures by evolving adjacency matrices through selection, crossover, and mutation. Fitness is evaluated via a task-specific loss, and the best individual is post-processed to produce the final graph. The population size is 50, the number of generations is 50, the mutation rate is 0.1, and the crossover rate is 0.8.

- **Simulated Annealing (SA)** This method uses Simulated Annealing to optimize graph structures by iteratively applying random perturbations and accepting changes based on a temperature-controlled probability. The number of iterations is 100, the initial temperature is 1.0, and the temperature decay rate is 0.99.

## E.2    CONTINUOUS RELAXATION

- **Acyclicity Handling** We follow the acyclicity characterization proposed in the NOTEARS framework Zheng et al. (2018), which expresses the DAG constraint as a smooth function:

$$h(P) := \text{tr}\left(e^{P \odot P}\right) - d,$$

  To enforce acyclicity during optimization, we adopt the quadratic penalty method by incorporating $h(W)^2$ into the objective function:

$$\min_P J(P) + h(P)^2,$$

- **Gumbel Softmax Sampling** We use the Gumbel-Softmax trick Jang et al. (2016) to obtain differentiable samples from a categorical distribution. By injecting Gumbel noise into the logits and applying a softmax with temperature $\tau = 1$, the resulting sample approximates a one-hot vector while remaining differentiable, enabling gradient-based optimization over discrete decisions. During training, the sampling batch size is 32 for an iteration with learning rate of 0.06.

- **Maximum Entropy Policy Sampling** We adopt the maximum entropy reinforcement learning framework Haarnoja et al. (2018), which encourages policies that are both reward-maximizing and stochastic. Specifically, the policy maximizes the expected return plus an

entropy regularization term, leading to exploration-aware behaviors:

$$\pi^* = \arg\max_{\pi} \mathbb{E}_{\pi}\left[\sum_t r_t + \alpha\mathcal{H}(\pi(\cdot|s_t))\right],$$

where $\alpha = 1.0$ controls the trade-off between reward and entropy. During training, the sampling batch size is 32 for an iteration with learning rate of 0.06.

### E.3 APPROXIMATION METHOD

**Held Routing** Held & Perner (2025) employs a modified Dijkstra algorithm that iteratively constructs routing trees by minimizing a hybrid cost-distance objective. The method dynamically adjusts edge weights during path exploration to balance construction cost and routing distance.

The algorithm iteratively constructs a Steiner tree via terminal merging, guided by a modified cost function. In each iteration, a pair of terminals $u, v$ minimizing

$$L(u,v) := \text{dist}_{G,c+\min\{w(u),w(v)\}\cdot d}(u,v) + b(u,v)$$

is selected, where $w(\cdot)$ denotes delay weights, $c(e)$ and $d(e)$ are edge cost and delay, and $b(u,v)$ is a bifurcation delay penalty. The algorithm uses sink-specific distance functions $\ell_u(e) := c(e) + w(u)\cdot d(e)$ to propagate labels in parallel from all sinks. After merging, a new Steiner vertex inherits delay weights and may serve as the root in the next iteration. This approach achieves an $O(\log|S|)$-approximation in $O(|S|(n\log n + m))$ time while accounting for bifurcation-induced delay penalties through a distributed model:

$$\text{delay}_T(r,t) = \sum_{(u,v)\in P}(d(u,v) + \lambda_v \cdot d_{\text{bif}})$$

where $\lambda_v \in [\eta, 1-\eta]$ is determined by subtree weights.The parameter settings are consistent with those in the original paper.

## F ADDITIONAL EXPERIMENTAL RESULTS

This section presents additional experimental results referenced in the main text.

Figure 3 complements the main text by presenting further experimental results comparing time cost and metric $J$ between Held and CDPG across four additional graphs. The x-axis ($|S'|$) indicates the number of randomly sampled source nodes, ranging from $0.5\log_2 n$ to proportional subsets of $|S|$. Overall, the trends align with those in Figure 2, with one notable exception: on Graph 9, CDPG achieves a performance metric $J$ nearly identical to Held's method, but at a significantly higher time cost. This discrepancy can be attributed to the small scale of Graph 9—the smallest among all datasets—where constant factors in practical computation become more pronounced, making this behavior expected.

Table 5: $J$ ($\times 10^4$) in different $\eta$ and $l$ in graph 3

| $\eta\backslash l$ | 2 | 4 | 6 | 8 | 10 | 12 |
|---|---|---|---|---|---|---|
| 0.02 | 24.88 | 25.14 | 24.80 | 24.67 | 24.72 | 24.73 |
| 0.06 | 24.93 | 24.74 | 24.76 | **24.63** | 24.77 | 24.83 |
| 0.10 | 24.82 | 24.87 | 24.98 | 24.83 | 24.78 | 24.85 |
| 0.14 | 24.80 | 24.98 | 25.05 | 24.96 | 24.81 | 25.11 |

## G USE OF LLM

We employed large language models (LLMs) solely for the purpose of improving grammatical accuracy and enhancing the clarity and rigor of language in this manuscript. All ideas, analyses, and

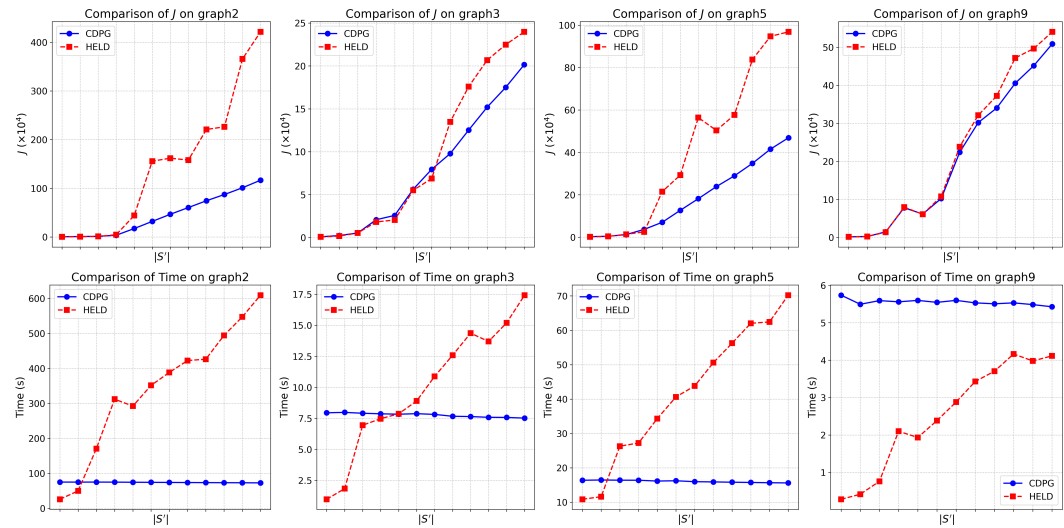

Figure 3: Comparison of time cost and metric $J$ between Held and CDPG across the other four graphs. The x-axis ($|S'|$) shows the count of randomly sampled source nodes, with values (left to right): $0.5\log_2 n$, $\log_2 n$, $4\log_2 n$, $16\log_2 n$, $0.1|S|$, $0.2|S|$, $\ldots$, $0.8|S|$.

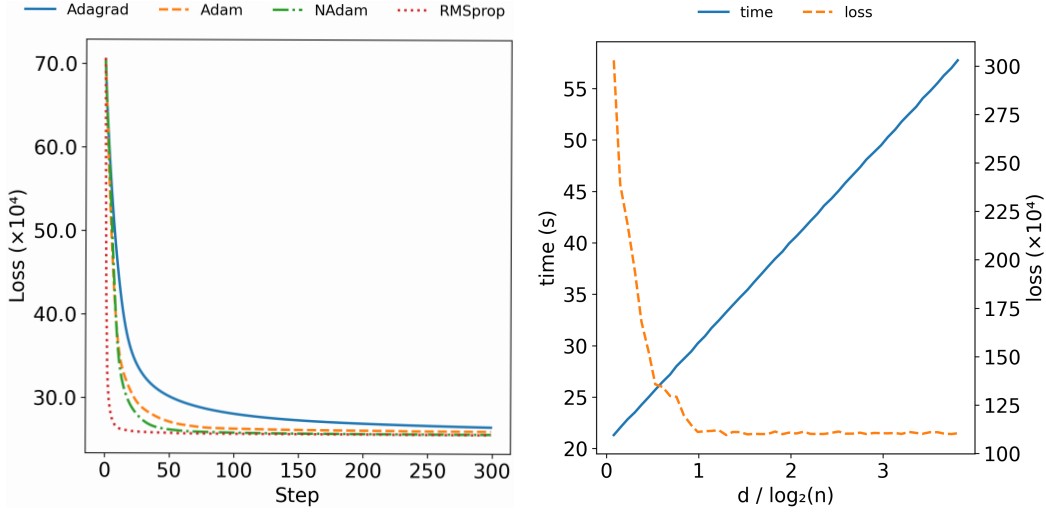

Figure 4: Loss Curve Comparison of Optimizers

Figure 5: Effect of $d$ on runtime and loss

conclusions remain entirely our own. The authors take full responsibility for all content presented in this work.

