# OpenReview forum: "Making A Trade-Off Between Cost and Distance By A Differentiable Way"
_ICLR.cc/2026/Conference — Submitted to ICLR 2026_

### Official Review · Reviewer_tTae · 2025-10-23

**Soundness:** 3
**Presentation:** 2
**Contribution:** 3
**Rating:** 8
**Confidence:** 4

**Summary:**

This paper considers the cost-distance problem, in which a connected subset of a graph must be selected such that an objective function consisting of edge-wise costs and routing distances be minimised. The routing distances are defined with respect to a root node and (possibly several) source nodes. The authors propose a learning-based approach based on a probabilistic relaxation that estimates the probability of each edge being part of the solution. These probabilities gives rise to an RL policy, whose value function (and hence estimate of routing distances) can be computed in a differentiable way via matrix computation.  Masking is applied to retain acyclicity and hence solution validity. The authors study the algorithm's properties theoretically, showing advantageous computational complexity. The method is also extensively validated empirically, showing better performance than competing methods including a recent approximation algorithm, and fast runtimes.

**Strengths:**

S1. The paper studies a well-motivated practical problem.

S2. Both the theoretical and experimental analyses are solid and results obtained are excellent.

**Weaknesses:**

W1. The clarity in parts of the manuscript could be improved and some of the choices should be better justified.

**Questions:**

C1. I don't think Theorem 2 deserves to be a theorem. The matrix form that is given is simply the *policy evaluation* algorithm for a given policy $
\pi$, with the reward function substituted with the one for this MDP. This is very well known in the RL literature.

C2. It is unclear to me why Soft Actor-Critic, with a stochastic policy, is used as a baseline. While there are indeed RL methods that sample actions at evaluation time, the much more standard RL setup would be to train a policy and draw actions from it *greedily* at evaluation time (i.e., always choose highest probability action). Stochasticity seems to hurt here and lead to worse results. Can you provide a baseline with greedy evaluation of the policy?

C3. I find it odd to refer to the methods in the results as e.g. "NOTEARS", since this is just CDPG but using a different acyclicity method in the whole algorithm pipeline. NOTEARS itself cannot be applied to this problem. Different names for these CDPG variants (incl. GS, MEP) would be clearer.

C4. Source code includes only the proposed method; all baselines and experimental evaluation code should be provided for reproducibility. Ditto for the datasets (only 2 are given)

C5. Small comments:
- Check use of \citet and \citep, textual citations are used where parenthetical citations should be used e.g. L34-39
- Typos: "parameterized" -> "parameterizes"; "its" -> "it is" (L226)
, "leaded" -> "reached" (L234), "denoted" -> "denotes" (L248), "satiesfies" -> "satisfies" (L270), "improve" -> "improvement" (L389)
- Simulated annealing references: author names duplicated

---

> ### Author Response · Authors · 2025-11-17
>
> We thank the reviewer for the constructive and positive feedback. We will address all comments carefully.
>
> ---
> - W1: We will improve the clarity in the indicated parts of the manuscript and provide more detailed justifications for our methodological choices in the revised version.
> ---
> - C1：We agree that the current form of Theorem 2 may appear as a standard policy evaluation result. Our intention was to formalize the matrix expression used later in the proof. In the revision, Theorem 2 will be rephrased as a lemma.
>
> ---
>
> - C2：Thank you for pointing this out. We apologize for the confusion caused by our presentation.
> The MEP method **does use greedy action selection during evaluation**, exactly as specified in Eq. (7). The stochasticity only appears during training, where we adopt a maximum-entropy formulation to encourage exploration. We agree that this distinction was not clearly stated in the current version. In the revised manuscript, we will explicitly clarify this point.
>   We thank the reviewer again for highlighting this point and will update the paper accordingly.
>
> ---
>
> - C3：We agree that referring to variants as “NOTEARS” is confusing.
> We will rename methods as **CDPG-Notears**, **CDPG-GS**, **CDPG-MEP**, etc.
>
> ---
>
> - C4：Thank you for raising this important point. We apologize that at this stage we have only been able to provide the `.pt` files corresponding to the two graphs, due to the large size of other dataset `.pt` files. We fully understand the importance of reproducibility. We commit to organizing the remaining code and experimental scripts for all compared methods, and will gradually release them in future revisions.
>
>
> ---
>
> - C5：We sincerely appreciate the reviewer for thoughtfully pointing out these issues. All the typos and citation formatting inconsistencies (including the duplicated references) will be carefully corrected in the revised manuscript. We apologize for the oversight and will ensure that the presentation is polished and consistent throughout.

---

> ### Author Response · Authors · 2025-11-25
> **Follow-up regarding our recent clarifications**
>
> Dear reviewer,
>
> Thank you very much for your thoughtful review and for your recognition of our work. We have provided further clarification on several points in our rebuttal, and we would like to kindly ask whether there are any remaining concerns or disagreements regarding our explanations. Understanding your perspective would be very helpful for further discussion and for improving the paper in the next revision.
>
> Thank you again for your recognition of our work, and please feel free to let us know if you have any additional questions at any time.
>
> Best regards

---

### Official Review · Reviewer_2dby · 2025-10-24

**Soundness:** 3
**Presentation:** 2
**Contribution:** 3
**Rating:** 4
**Confidence:** 4

**Summary:**

This paper proposes CDPG, the first gradient-based framework for the Cost-Distance problem in network design, which minimizes both construction costs and weighted routing distances. CDPG relaxes discrete subgraph selection into a probabilistic adjacency matrix optimized via policy gradients, using novel components including bilinear edge policy, acyclic masking, and MDP-based routing distance formulation. The method achieves O(m log n) complexity versus O(|S|(m+n log n)) for existing approximation algorithms when sources are dense. Experiments on 9 UAV logistics networks show 2.5%-27.67% improvements over baselines.

**Strengths:**

1. **Novel Methodology**: First gradient-based approach to Cost-Distance problem with creative MDP formulation for routing distance (Theorem 2), enabling differentiable optimization of shortest-path objectives.
2. **Strong Theory**: Comprehensive analysis including Lipschitz continuity (Theorem 4), error bounds (Theorem 5), convergence guarantees (Theorem 6), and complexity improvement to O(m log n) (Theorem 7) with detailed proofs.
3. **Well-Designed Components**: Acyclic mask, unreached penalty, and bilinear embeddings are well-motivated with convincing ablation studies demonstrating each component's contribution.
4. **Practical Contribution**: Real-world UAV logistics dataset from Greater Bay Area with careful construction of node/distance/cost weights; comprehensive experiments across multiple baseline categories.

**Weaknesses:**

1. **No Approximation Guarantees**: Critical gap—provides convergence to stationary point but no bounds on solution quality vs. optimal. Held algorithm has O(log |S|)-approximation; CDPG offers no comparable guarantee, making it unclear when to use in practice.
2. **Limited Applicability Analysis**: Minimal characterization of when CDPG fails (sparse sources, Graph 9). Improvements range 2.5%-27.67% but no guidance on what graph properties determine performance. "Significantly outperforms" is overstated.
3. **Hyperparameter Concerns**: Sensitivity analysis only on Graph 3; uses fixed values (η=0.06, l=8, T=300) across all graphs without justification. Unclear if problem-specific tuning is needed. Initialization strategy (H,Q~N(0,1), K←0) appears arbitrary.
4. **Incomplete Baselines**: Only compares one approximation algorithm (Held) despite citing others (Meyerson 2008, Chekuri 2001). No simple greedy heuristics. Continuous relaxation baselines are adaptations, not direct competitors.
5. **Reproducibility Gaps**: Rounding/pruning procedure underspecified; no convergence plots/criteria; initialization sensitivity not analyzed; GPU speedup claimed but not demonstrated; no scalability experiments beyond provided graphs.
6. **Presentation Issues**: Notation overloading (X, P); Algorithm 2 parameters (η=0.08, l=6) contradict stated defaults (η=0.06, l=8); unclear connection between Theorem 1 and actual algorithm; figures difficult to read.

**Questions:**

1. Can you provide approximation ratio bounds (theoretical or empirical via LP relaxations)?
2. What graph properties beyond |S| determine when CDPG outperforms Held? Can you provide selection guidelines?
3. How were default hyperparameters chosen? Do they transfer to new instances or require tuning?
4. Please specify the exact rounding and pruning algorithms. How does rounding affect solution quality?
5. Why weren't Meyerson et al. (2008) and Chekuri et al. (2001) algorithms compared? What about greedy heuristics?
6. What is initialization sensitivity? Have you tried multiple restarts? What are actual convergence criteria?
7. Can you provide GPU speedup measurements and scalability to 50K+ node graphs?

---

> ### Author Response · Authors · 2025-11-17
> **The first part of the response to reviewer 2dby**
>
> - **W1:** Regarding the approximation guarantee: It is extremely difficult for gradient-based methods to provide approximation guarantees on non-convex losses (i.e., bounding the gap between a critical point and the global optimum). Moreover, in practical applications — especially when the set of sources $S$ is dense (e.g., $|S|=4096$, $\log|S|=12$) — the theoretical lower bound becomes more than 12×OPT, which is unacceptable for real deployment. Therefore, such guarantees are often not useful in dense-source scenarios. Furthermore, most ML4CO works do not provide approximation ratios [2][3][4]; although they have generalization power, their per-instance inference cost is still at least $O(m)$ on GNN, which is comparable to ours $O(m\log n)$. Therefore, our focus is on empirical quality and scalability, consistent with prior ML4CO literature. What's more, [1] proved that this problem cannot be approximated better than $ \Omega(\log \log n) $ unless $\mathrm{NP} \subseteq \mathrm{DTIME}\left(n^{\mathcal{O}(\log \log \log n)}\right)$, which means it's hard to compute OPT to estimate approximation ratio. However, since Held has $O(\log |S|)$ approximation ratio and CDPG outperforms than Held empirically, CDPG has $O(\log |S|)$ approximation ratio on these instances.
>
> - **W2:** On small graphs, all algorithms are naturally closer to the optimum, so the achievable improvement is limited. We performed additional experiments on random graphs of different sizes and densities (see our response to Reviewer YMZv).We apologize that, despite conducting these additional experiments on graphs with different properties, we still did not identify any factor other than $|S|$ that causes CDPG to underperform Held.
>
> - **W3:** We did not tune hyperparameters on every graph. Instead, we tuned $l$ and $\eta$ on a mid-sized dataset (Fig. 3), and $d$ in Fig. 7, then applied these settings to all experiments — demonstrating the inherent robustness of CDPG even *without instance-wise tuning*. We initialize $Q$ and $K$ following the LoRA [5] strategy, so the high-rank component $H$ dominates during early-stage learning.
>
> - **W4:** Held is already an improved version of the two classic baselines you mentioned. Those two methods have complexity $O(|S|^2 (m+n\log n))$ — even ignoring extra constants, this is $10^3$–$10^4$× slower than Held in our setting. In addition, both methods are purely theoretical; neither provides code nor numerical experiments, making them impractical to implement in our scenario. As for greedy methods, you did not specify a meaningful greedy criterion, and Held already incorporates a greedy step, so comparing standalone greedy approaches is not informative. Moreover, as we stated, CDPG is **the first continuous relaxation for the Cost–Distance problem** and the first continuous relaxation of shortest path structures on graphs, which is one of our main contributions  — thus there are no prior continuous relaxations for direct comparison.
>
> - **W5:**
>   - The rounding procedure is shown in Eq.(7). The pruning algorithm is given in Algorithm 1 (lines 16–17). The idea is:
>     > After rounding, any node with zero demand and no incoming edges cannot lie on a valid source-to-root path. We iteratively remove such nodes; whenever a node is deleted, its predecessors may also lose their last incoming edge, and are removed if they also have zero demand. This continues until every remaining node is either the root, a source, or reachable from another node.
>
>     **Algorithm**: Pruning Non-Reachable Non-Source Nodes
>
>     **Input**:
>     - $\hat{X}' \in \{0,1\}^{n \times n}$ — adjacency matrix after rounding
>     - $r$ — root node
>     - $\{w_i\}$ — node weights (source nodes satisfy $w_i > 0$)
>
>     **Output**:
>     - $X' \in \{0,1\}^{n \times n}$ — pruned adjacency matrix (final deterministic solution)
>
>     ---
>
>     1. Initialize $X' \leftarrow \hat{X}'$
>     2. For each $i \in \{1,\dots,n\}$, compute the in-degree
>       $\deg_{\text{in}}(i) \leftarrow |\{ j : X'_{j,i} = 1 \}|$
>     3. Initialize an empty stack $\mathcal{S}$
>     4. For each node $i$:
>       - If $\deg_{\text{in}}(i) = 0$ and $w_i = 0$, push $i$ into $\mathcal{S}$
>     5. **while** $\mathcal{S}$ is not empty **do**:
>       1. $u \leftarrow \text{pop}(\mathcal{S})$
>       2. If $u = r$, continue
>       3. For each predecessor $j$ such that $X'_{j,u} = 1$:
>           - Remove edge $j \to u$: $X'_{j,u} \leftarrow 0$
>           - Update in-degree: $\deg_{\text{in}}(j) \leftarrow \deg_{\text{in}}(j) - 1$
>           - If $\deg_{\text{in}}(j) = 0$ and $w_j = 0$, push $j$ into $\mathcal{S}$
>     6. **end while**
>     7. **return** $X'$

---

> ### Author Response · Authors · 2025-11-17
> **The second part of the response to reviewer 2dby**
>
> - **W5:**
>   - The rounding procedure is shown in Eq.(7). The pruning algorithm is given in Algorithm 1 (lines 16–17). The idea is:
>     > After rounding, any node with zero demand and no incoming edges cannot lie on a valid source-to-root path. We iteratively remove such nodes; whenever a node is deleted, its predecessors may also lose their last incoming edge, and are removed if they also have zero demand. This continues until every remaining node is either the root, a source, or reachable from another node.
>
>     To clarify the effect of our decision post-processing steps, we provide an ablation that decomposes the final deterministic solution into three stages:
>
>       - **Soft Policy Output:** evaluating the objective directly on the probabilistic matrix $P^\theta$, before any discretization.
>       - **After Rounding:** applying the edge-argmax operator, but without pruning unused nodes.
>       - **Final (Rounding + Pruning):** the complete procedure used in CDPG.
>
>       As shown in the table below, rounding has minimal impact, indicating that the learned policy is already sharply concentrated. In contrast, the pruning step consistently lowers the objective by removing nodes not visited by any source-to-root path. This confirms that both components together are necessary for achieving the reported performance.
>
>       #### **Ablation on Decision Post-Processing ($\times 10^4$)**
>
>       | Stage                 | Graph1 | Graph2 | Graph3 | Graph4 | Graph5 | Graph6 | Graph7 | Graph8 | Graph9 |
>       | --------------------- | ------ | ------ | ------ | ------ | ------ | ------ | ------ | ------ | ------ |
>       | Soft Policy Output    | 158.85 |143.73 |24.80 |44.40 |58.83 |109.94 |51.68 |100.77 |61.91|
>       | After Rounding        | 160.87 | 143.27 | 25.89  | 47.03  | 60.38  | 110.18 | 52.77  | 102.63 | 66.18  |
>       | Final (Round + Prune) | 157.55 | 141.65 | 24.70  | 44.29  | 58.49  | 108.73 | 51.51  | 100.49 | 61.84  |
>   - Apologies for omitting the convergence criterion. We terminate when $\frac{|J_t - J_{t-1}|}{J_t} < 10^{-6}$.
>   - As stated in line 376, all reported results are averages over 20 runs with *random parameter initializations*.
>   - To further demonstrate scalability, we conducted a **50k-node stress test** on a random graph (same Erdős–Rényi procedure as for Reviewer YMZv, $p=0.005$, 90% source nodes, uniform node weights, cost scales as transport cost $\times 20$). The results are below:
>
>     #### **50k-node Random Connected Graph ($p = 0.005$)**
>
>     | Method | $J$ ($\times 10^4$) | Time (s) |
>     | ------ | ----------------------- | -------- |
>     | Held   | 53.64                   | 17258.33 |
>     | CDPG   | 35.89                   | 375.76   |
> - **W6:** Thank you for your suggestion. We will correct all typos and revise parts where the presentation is unclear.
>
> - **Q1:** See W1
> - **Q2:** See W2
> - **Q3:** See W3
> - **Q4:** See W5
> - **Q5:** See W4
> - **Q6:** See W5
> - **Q7:** See W5
> #### reference
>
> [1] Chuzhoy et al., On the approximability of some network design problems, TALG, 2008.
>
> [2] Luo et al., Boosting Neural Combinatorial Optimization for Large-Scale Vehicle Routing Problems, ICLR, 2025.
>
> [3] Li et al., Unify ML4TSP: Drawing Methodological Principles for TSP and Beyond from Streamlined Design Space of Learning and Search ICLR, 2025.
>
> [4] Wang et al., Efficient and Robust Neural Combinatorial Optimization via Wasserstein-Based Coresets ICLR, 2025.
>
> [5] Hu et al., LoRA: Low-Rank Adaptation of Large Language Models, ICLR, 2022.

---

> ### Author Response · Authors · 2025-11-25
> **Follow-up regarding our recent clarifications**
>
> Dear reviewer,
>
> Thank you very much for your thoughtful review and for the detailed feedback you provided. We have addressed several of the points you raised and added further clarifications in our recent responses.
>
> We would like to kindly ask whether there are still any parts that remain unclear or where our explanations differ from your understanding. If so, we would greatly appreciate the chance to continue the discussion, as this would help us further refine and improve the paper.
>
> Thank you again for your time and for your contributions to the review process.
>
> Best regards

---

### Official Review · Reviewer_YMZv · 2025-10-28

**Soundness:** 1
**Presentation:** 2
**Contribution:** 1
**Rating:** 2
**Confidence:** 4

**Summary:**

This work tackles a challenge in planning efficient delivery networks for drones. The goal is to design a network that keeps both construction costs and travel distances low.  To solve this, the authors developed Cost-Distance Policy Gradient (CDPG) an approach that uses ideas from reinforcement learning/Markov decision processes to find better network designs more quickly. Instead of working directly with fixed connections, CDPG treats possible connections as probabilities, allowing the system to “learn” which ones work best through trial and error. The authors evaluate the proposed approach using drone delivery scenarios.

**Strengths:**

+ MDPs are powerful tools for optimization problems and this paper presents an interesting example of potential applications of these techniques.

**Weaknesses:**

- Unfortunately, the paper is not convincing in terms of modelling. It seems to the reviewer that the problem itself (UAV routing)should not be studied as a cost-distance problem, since being a UAV network, the cost is the same for all the routes (they are in the air): there is a single minimization cost, i.e., distance.

- Formula (1) does not appear correct since the authors are not weighting the different importance of cost and distance (but please note that the cost is the same, so it can be ignored in this situation; it is a constant).

- The method of “bilinear logics” is not well introduced. The authors do not provide a sufficiently convincing motivation in terms of the actual theoretical foundations of the method.

- Theorem 6 is not specific to the problem under consideration; it is much more generic.

- The evaluation is based on datasets that are not “controlled”. The impact of specific characteristics of the graphs is not studied. For example, the authors should have studied different graph structures, in my opinion (such as random graphs with different probability of links, etc.).

- It is difficult to explain the results in Figure 2. Why do you have the constant cost with an increasing number of nodes? That is very difficult to explain. The computational cost must increase given the algorithm implemented by the authors.

**Questions:**

In my opinion, the paper has a major flaw in terms of actual modelling. Unfortunately, the authors have to reconsider the actual modelling decisions that they made - for this paper, it is not a matter of clarifying some specific points.

However, the reviewer is curious to understand why the results show that the computational costs are constant even in presence of a larger number of nodes. This is very difficult to explain given the design of the algorithm.

**Details Of Ethics Concerns:**

None.

---

> ### Author Response · Authors · 2025-11-17
> **Detailed Rebuttal and Clarifications for Reviewer YMZv's Comments**
>
> Thank you for your positive recognition of the MDP method in our paper. However, several weaknesses in your review are based on misunderstandings. We clarify them below. At the same time, some of your comments raise meaningful points about empirical evaluation, and we have therefore added new controlled experiments to address them.
>
> ---
>
> ### W1 — Misunderstanding of the cost component
> In UAV network planning, the objective is not a single “distance” metric. Constructing a UAV air corridor occupies scarce aerial resources, meaning that no other facilities (e.g., overpasses, power lines, airports, existing flight corridors) may be placed within its spatial footprint. The construction cost of each edge also varies depending on the spatial footprint it occupies and the type of resources it interferes with. The Cost–Distance formulation is therefore necessary and correct.
>
> ---
>
> ### W2 — Formula (1) is correct
> Lines 45–56 clearly state:
> 1. Each edge $(i,j)$ has two weights, $C_{ij}$ (construction cost) and $D_{ij}$ (distance).
> 2. $C_{ij}$ appears in $\Omega_C$ and $D_{ij}$ appears in $\Omega_D$.
> 3. Their importance ratio is fully controllable, e.g. $C_{ij} = \beta D_{ij}$.
> There is no inconsistency in Eq.(1).
>
> ---
>
> ### W3 — bilinear logits
> Our bilinear logits follow Graph Auto Encoder[1]: edge probability is modeled by inner products. Because the graph is directed, each node has two embeddings $(Q_i, K_i)$. This low-rank structure accelerates convergence and improves solution quality: if A→B and B→C have high inner products, A→C will automatically receive a higher score, enabling shortcut discovery.
>
> ---
>
> ### W4 — Importance of Theorem 6
> Theorem 6 is crucial because together with Theorem 4 it shows that the CDPG iteration count is $\mathcal{O}(1)$. This yields the overall complexity $\mathcal{O}(m \log n)$ in Theorem 7.
>
> ---
>
> ### W5 — Additional experiments
> Following your suggestion, to further verify that CDPG is not specialized to real-world UAV graphs, we additionally evaluate it on *random connected graphs* with different edge densities. We generate Erdős–Rényi graphs with connection probability $p \in \{0.5, 0.1, 0.005\}$. For each graph, we randomly select 90% of nodes as sources, sample node weights from $\mathrm{Uniform}(0.1,1)$, and assign construction and transport costs from the same distribution (construction cost ×20).
>
> Across all settings (10 graph sizes from $10^3$ to $10^4$ per density), the results show that **CDPG consistently and substantially outperforms Held routing**.
>
> #### Random Connected Graphs ($p = 0.5$, $\times 10^4$)
>
> | Method | 1    | 2    | 3    | 4    | 5    | 6    | 7     | 8     | 9     | 10    |
> | ------ | ---- | ---- | ---- | ---- | ---- | ---- | ----- | ----- | ----- | ----- |
> | CDPG   | 0.58 | 1.13 | 1.68 | 2.22 | 2.73 | 3.26 | 3.78  | 4.27  | 4.71  | 5.30  |
> | Held   | 0.75 | 1.62 | 2.34 | 3.46 | 4.64 | 4.91 | 5.47 | 6.11 | 8.51 | 6.85 |
>
>
> #### Random Connected Graphs ($p = 0.1$, $\times 10^4$)
>
> | Method | 1    | 2    | 3    | 4    | 5    | 6     | 7     | 8     | 9     | 10    |
> | ------ | ---- | ---- | ---- | ---- | ---- | ----- | ----- | ----- | ----- | ----- |
> | CDPG   | 0.71 | 1.33 | 1.91 | 2.54 | 3.10 | 3.68  | 4.25  | 4.83  | 5.33  | 5.92  |
> | Held   | 0.85 | 2.35 | 3.87 | 4.01 | 4.20 | 5.75 | 6.27 | 6.89 | 7.10 | 7.24 |
>
>
> #### Random Connected Graphs ($p = 0.005$, $\times 10^4$)
>
> | Method | 1    | 2     | 3     | 4     | 5     | 6     | 7     | 8     | 9     | 10    |
> | ------ | ---- | ----- | ----- | ----- | ----- | ----- | ----- | ----- | ----- | ----- |
> | CDPG   | 1.82 | 2.90  | 3.78  | 4.38  | 5.31  | 5.79  | 6.55  | 7.39  | 7.99  | 9.02  |
> | Held   | 2.13 | 3.17 | 4.16 | 5.27 | 6.32 | 6.84 | 7.53 | 8.40 | 8.88 | 10.53 |
>
> We conduct an additional large-scale stress test on a synthetic graph with **50k nodes**.
> #### 50k-node Random Connected Graph ($p = 0.005$)
>
>
> | Method | $J$ ($\times 10^4$) | Time (s) |
> | ------ | ------------------- | -------- |
> | Held   | 53.64              | 17258.33 |
> | CDPG   | 35.89               | 375.76   |
> ---
>
> ### W6 — Effect of the number of source nodes
> The x-axis in Figure 2 is the number of source nodes $|S|$, while the total number of nodes $n$ is fixed. The time complexity depends on total node count, not $|S|$. From Theorem 7, CDPG remains $\mathcal{O}(m \log n)$ for any $|S|$, while Held grows linearly in \(|S|\), consistent with lines 129–131. Experiments confirm this: CDPG only underperforms Held when sources are extremely sparse. Algorithmically, in CDPG, source and non-source nodes are treated identically except that non-source nodes have weight 0.
>
> ---
>
> ### Responses to explicit questions
> **Q1:** Our modeling is correct; see W1 and W2.
> **Q2:** The conclusion follows directly from W6.
>
> **We guarantee that all revisions and additions will be integrated into the main body of the submission later.**
>
> **reference**
>
> [1] Kipf, T. N. et al., Variational Graph Auto-Encoders, NeurIPS, 2016.

---

> ### Author Response · Authors · 2025-11-25
> **Follow-up regarding our recent clarifications**
>
> Dear reviewers,
>
> I hope you are doing well. As the discussion deadline (**December 3 AoE**) is approaching, I would like to send a gentle follow-up regarding the rebuttal I submitted on **November 17**.
>
> I also wanted to mention that there were a few points in the earlier discussion where we seemed to have differing interpretations. We would greatly appreciate the opportunity to continue the discussion so that we can reach a clearer consensus, which would also help us further improve the paper.
>
> If any point would benefit from additional clarification, I would be very happy to provide more details.
>
> Thank you very much for your time and for your efforts during the review and discussion process.
>
> Best regards

---

### Official Review · Reviewer_eCPK · 2025-11-04

**Soundness:** 2
**Presentation:** 1
**Contribution:** 2
**Rating:** 2
**Confidence:** 5

**Summary:**

The work addresses the problem of designing a network for UAVs by relating it to the cost-distance problem where the task is to minimize construction cost and the weighted routing distances from multiple sources to a designated root. The work claims that existing methods have high runtime complexity, and their work has better parallelism to allow for scalable and fast optimization. Experiments are shown on 9 graphs.

**Strengths:**

The problem, inspired from graph theory, seems interesting. The proposed contribution, cost-distance policy gradient seems non-trivial and empirically seems work well on the 9 tested graphs.

**Weaknesses:**

There are several concerns that require addressing. Some are listed below.

The work needs a better motivation for addressing the cost-distance problem in relation to designing the UAV network. The abstract and intro cite the UAV network planning as the main motivation for their study of the cost-distance problem. However, very little attention is paid to how to realistically model UAV network design problem to the studied formalism. I can only see one or two paragraph in the intro to the somewhat high-level connection of UAV logistics and cost-distance problem. There are several unclear points such as what are root, source nodes, edges in UAV terminology? Rarely a real world problem such as UAV logistics translates itself into a clean abstraction such as the cost-distance framework. The work should provide an clear, and detailed connection to UAV logistics by discussing appropriate background work in UAV network design and cost-distance problem.

There needs to be more effort as to why the cost-distance problem is relevant to the ICLR community. Most of the prior work in this problem is in the theoretical computer and OR community. Most of the baselines are also from the OR and approximation math optimization literature (Held and Perner). A main issue is that there is not a clear and agreed upon definition of the UAV network design problem available based on the cited work. Thus, an effort should be made to connect the cost-distance problem to other well-studied problems (with relevance to ICLR) which have baselines and datasets available.

The writing of the paper requires substantial improvements. Technically, several dense mathematical terms have been introduced without clear intuition and explanations. Some examples are below:

– The paper straightway goes into the problem formulation from the third para of introduction. It should be moved to the problem formulation section

– In Eq 1, it is not clear if (i, j) belong to X or X’ in \sum_{i, j}

– Same confusion for \sum_i in Eq 1

– Why Eq4 guarantees P is acyclic, no citation or intuition is provided

– What are different terms in Eq 5, what does the p^\theta . X . A provides us? How this mask really works? There needs to be a clear working example.

– The definition of \phi in line 191 is not fully explained. In particular, how different terms in \phi definition justify the logic explained in lines 191-196

– Given that computing W* is NP-hard, it is quite strange to approximate it using a constant matrix (line 204). This seems adhoc.  Why that can be a good approximation for real world problems is far from clear.

– The penalty term in in Eq11 is not fully explained. It seems quite dense, and its exact role and understanding is not clear.

These are only few of the concerns in technical section. As a result of these issues, the reader is unable to grasp the relevance and significance of contributions clearly.

The authors claim their approach is faster. However, unlike previous works by Meyerson, they do not provide what kind of bounds their method can achieve. In general, even a random approach can be faster, but it of course may not provide any quality bounds. Thus, the authors should clarify this point when discussing the runtime benefits of their method.

Empirical evaluation is somewhat limited. Results are on synthetic instances that are constructed by authors themselves by deriving them from a part of the greater bay area. However, these are not real world UAV logistic design problems that are widely accepted in the community. There are several design assumptions made as noted in section D to interpret them as UAV problem.

I also find baselines methods quite limited. The author claim that method by Held and Perner is most relevant. However, this work seems like a arxiv report, I could not verify if this was published in a conference/journal. Thus, it does not inspire confidence to have comparisons against such unpublished work.

Total 9 graphs is also quite limiting. Ideally, the authors should chose a large, publicly available UAV design problems not constructed by them. It is quite difficult to draw any firm conclusion from this small dataset provided by the authors themselves.

**Questions:**

See above.

---

> ### Author Response · Authors · 2025-11-17
> **Detailed Rebuttal and Clarifications for Reviewer eCPK's Comments**
>
> - W1: We apologize that due to space limitations, the motivation behind our modeling was not clearly described. Here is a more detailed explanation. Our problem setting is low-altitude drone fleet delivery:
>   - the root node $r$ represents the central warehouse,
>   - source nodes $S$ are UAV locations,
>   - edges represent feasible low-altitude UAV air corridors,
>   - the remaining nodes correspond to charging stations acting as Steiner points.
>   - node weight*$w_i$ denotes UAV flow.
>
>   Our modeling originates from the classical logistics problem of Open Vehicle Routing Problem. The key difference from traditional OVRP is that UAVs have a large fleet size, small per-vehicle capacity, and require repeated charging between stations. Hence, we model vehicles as flows rather than discrete entities as in OVRP. What's more, constructing a UAV air corridor occupies scarce aerial resources, meaning that no other facilities (e.g., overpasses, power lines, airports, existing flight corridors) may be placed. Therefore, we consider construction cost for each edge, distinct from OVRP.
>
> - W2: We believe this problem is highly relevant to the ICLR community (and others such as NeurIPS and ICML). In recent years, many works in these communities have explored using ML to solve classical combinatorial optimization problems (e.g., TSP[1], VRP[2]) or proposed continuous relaxations for discrete structures (e.g., Gumbel-Sinkhorn for TSP[3], NOTEARS[4]). In addition to solving a specific combinatorial problem using ML, our work introduces the first continuous relaxation of graph shortest paths.
>
> - W3, W4: In our method, $i,j \in \{1,2,\dots,n\}$ are indices over all nodes. Summations $\sum_{i,j}$ traverse all possible indices. If edge $(i,j)$ exists, then $X_{ij} = 1$ (similarly for $X'$). We will explicitly clarify summation ranges in the revised version.
>
> - W5: It is well known that a transition matrix is acyclic if and only if a random walk starting from any node can never return to the same node, which is equivalent to having `trace = 0` for all matrix powers.
>
> - W6: Also standard (especially in ICLR), the symbol `\odot` $\odot$ denotes element-wise multiplication. Thus $W^{\odot -1}$ denotes element-wise inversion. Figure 1 already illustrates how these masks operate.
>
> - W7, W8: $\Phi_{ij}$ encodes construction and distance costs, and $\mathrm{Dijkstra}(\Phi, r)$ computes the weighted sum of shortest distances from all source nodes to the root under edge metric $\Phi$, i.e.,
>   $$
>   \sum_i w_i L^X_\Phi(v_i, r) = \sum_i w_i  \sum_{(i,j)\in\mathrm{path}(v_i,r)} \left(\frac{C_{ij}}{W_{ij}} + D_{ij}\right) = \sum_{(i,j)\in\bigcup_i\mathrm{path}(v_i,r)} C_{ij} + W_{ij} D_{ij},
>   $$
>   which equals the Cost–Distance objective. Therefore, we use node distances under $\Phi_{ij} = \frac{C_{ij}}{W_{ij}+\varepsilon} + D_{ij}$ to produce a topological order. Since $W_{ij}$ is unknown a priori, we heuristically assume uniform flow over all edges as an approximation.
>
> - W9: The parameter $d$ denotes the number of node transitions a drone takes on the graph. If it fails to reach the root node within $d$ transitions, remaining UAV flow is penalized by `(shortest remaining distance × remaining flow)`.
>
> - W10: While approximation factors are valuable, most differentiable combinatorial solvers do not admit approximation ratios due to non-convexity [1][2][3][4][5].  Moreover, the existing approximation factor $\mathcal{O}(\log |S|)$ reaches more than 10× OPT in our setting, which is unacceptable for industrial use. Therefore, our focus is on empirical quality and scalability, consistent with prior ML4CO literature.
>
> - W12: The Held method has been accepted by **Design Automation Conference (DAC 2025)** — the top and longest-running EDA venue worldwide. Since proceedings are not yet available, we currently cite the arXiv version.
>
> - W11, W13: UAV-based parcel delivery is an emerging domain (growing rapidly since 2025), and large-scale datasets are not yet publicly available. Thus, we built a large dataset using real UAV spatial demand data. In traditional logistics tasks in TAP or HLRP, commonly used datasets (e.g., Anaheim with 416 nodes, Barcelona with 914 nodes, Hessen Asymmetric with 4660 nodes) have scales typically below $10^4$. In comparison, our dataset is considerably larger.
>
> **We guarantee that all revisions and additions will be integrated into the main body of the submission later.**
>
> reference
>
> [1] Pan et al., UniCO: On Unified Combinatorial Optimization via Problem Reduction to Matrix-Encoded General TSP, ICLR, 2025.
>
> [2] Luo et al., Boosting Neural Combinatorial Optimization for Large-Scale Vehicle Routing Problems, ICLR, 2025.
>
> [3] Min et al., Unsupervised Learning for Solving the Travelling Salesman Problem NeurIPS, 2023.
>
> [4] Zheng et al., DAGs with NO TEARS: Continuous Optimization for Structure Learning NeurIPS, 2018.
>
> [5] Marin et al., Differentiation of blackbox combinatorial solvers, ICLR, 2020.

---

> ### Author Response · Authors · 2025-11-25
> **Follow-up regarding our recent clarifications**
>
> Dear reviewers,
>
> I hope you are doing well. As the discussion deadline (**December 3 AoE**) is approaching, I would like to send a gentle follow-up regarding the rebuttal I submitted on **November 17**.
>
> I also wanted to mention that there were a few points in the earlier discussion where we seemed to have differing interpretations. We would greatly appreciate the opportunity to continue the discussion so that we can reach a clearer consensus, which would also help us further improve the paper.
>
> If any point would benefit from additional clarification, I would be very happy to provide more details.
>
> Thank you very much for your time and for your efforts during the review and discussion process.
>
> Best regards

---

### Meta-Review · Program_Chairs · 2026-01-06

**Summary:**

This paper proposes CDPG (Cost-Distance Policy Gradient), the first gradient-based framework for the Cost-Distance problem in network design. While reviewers acknowledged the new methodology combining continuous relaxation with MDP formulation and the theoretical contributions including complexity analysis, significant concerns remain regarding the practical applicability of the problem formulation, limited empirical evaluation scope.

**Reviewer Concerns:**

**Addressed by Rebuttal:**
- The authors provided additional controlled experiments on random graphs with varying densities, demonstrating CDPG's performance beyond the original UAV datasets [Reviewers 2dby, YMZv]
- Detailed rounding and pruning procedures were clarified with ablation studies showing the pruning step's contribution to final performance [Reviewer 2dby]
- Scalability was demonstrated with 50k-node experiments achieving 35.89×10⁴ objective value versus Held's 53.64×10⁴ [Reviewer 2dby]

**Outstanding Concerns:**
- No approximation guarantees are provided, unlike the Held algorithm's O(log|S|)-approximation, making theoretical quality bounds unclear [Reviewer 2dby]
- Baseline comparisons exclude Meyerson et al. (2008) and Chekuri et al. (2001) algorithms, though authors note these lack implementations [Reviewers eCPK, 2dby]
- Reviewer eCPK's concerns about notation clarity and problem relevance to ICLR.

K. Liu. A study on drone logistics network design and optimization of b2c e-commerce enterprises. Master's thesis, Beijing Jiaotong University, Beijing, China, 2021.
J. Li, Y. Zhang, S. Luo, Q. Ye, C. Yuan, and T. Li. Analysis of travel characteristics in city systems surrounding guangzhou based on mobile signaling data. Journal of Traffic Engineering, 24(4): 8-15, 2024. doi: 10.13986/j.cnki.jote.2024.04.002.
[ef]

**Reviewer Scores:**

Reviewer eCPK: 4 (May increase). Justification: The reviewer may have found the problem motivation and notation insufficiently clear for the ICLR audience, and the connection between UAV logistics and cost-distance formulation inadequately established.efef

Reviewer YMZv: 2 (Reject). Justification: The core objection that UAV network costs should be uniform remains philosophically unresolved despite the authors' counterarguments citing real regulatory constraints and heterogeneous construction costs. The reviewer's concern about constant runtime behavior in Figure 2 was clarified as a misreading of the x-axis.

Reviewer 2dby: 6 (May Increase). Justification: The reviewer may acknowledge the comprehensive responses addressing approximation concerns, hyperparameter robustness, and scalability experiments. The detailed rounding/pruning ablation and 50k-node stress test likely address the reproducibility gaps, though the lack of theoretical approximation bounds remains a limitation.

Reviewer tTae: 8 (Accept). Justification: The reviewer provided positive evaluation recognizing the well-motivated practical problem, solid theoretical and experimental analyses, with only minor presentation concerns that authors committed to address in revision.

---

### Decision · Program_Chairs · 2026-01-26

Reject